



# Forward and inverse modeling of water flow in unsaturated soils with discontinuous hydraulic conductivities using physics-informed neural networks with domain decomposition

Toshiyuki Bandai[1] and Teamrat A. Ghezzehei[1]

[1]Life and Environemental Science Department, University of California, Merced, Merced, CA, USA

**Correspondence:** Toshiyuki Bandai (tbandai@ucmerced.edu)

**Abstract.** Modeling water flow in unsaturated soils is vital for describing various hydrological and ecological phenomena. Soil water dynamics is described by well-established physical laws (Richardson-Richards equation (RRE)). Solving the RRE is difficult due to the inherent non-linearity of the processes, and various numerical methods have been proposed to solve the issue. However, applying the methods to practical situations is very challenging because they require well-defined initial and boundary conditions. Recent advances in machine learning and the growing availability of soil moisture data provide new opportunities for addressing the lingering challenges. Specifically, physics-informed machine learning allows taking advantage of both the known physics and data-driven modeling. Here, we present a physics-informed neural networks (PINNs) method that approximates the solution to the RRE using neural networks while concurrently matching available soil moisture data. Although the ability of PINNs to solve partial differential equations, including the RRE, has been demonstrated previously, its potential applications and limitations are not fully known. This study conducted a comprehensive analysis of PINNs and carefully tested the accuracy of the solutions by comparing them with analytical solutions and accepted traditional numerical solutions. We demonstrated that the solutions by PINNs with adaptive activation functions are comparable with those by traditional methods. Furthermore, while a single neural network (NN) is adequate to represent a homogeneous soil, we showed that soil moisture dynamics in layered soils with discontinuous hydraulic conductivities are correctly simulated by PINNs with domain decomposition (using separate NNs for each unique layer). A key advantage of PINNs is the absence of the strict requirement for precisely prescribed initial and boundary conditions. In addition, unlike traditional numerical methods, PINNs provide an inverse solution without repeatedly solving the forward problem. We demonstrated the application of these advantages by successfully simulating infiltration and redistribution constrained by sparse soil moisture measurements. As a free by-product, we gain knowledge of the water flux over the entire flow domain, including the unspecified upper and bottom boundary conditions. Nevertheless, there remain challenges that require further development. Chiefly, PINNs are sensitive to the initialization of NNs and are significantly slower than traditional numerical methods.

## 1 Introduction

Near-surface soil moisture is a critical variable for understanding land-atmosphere interactions. Its applications range from hydrological modeling and agricultural water management to the prediction of natural disasters (Robinson et al., 2008; Vereecken





et al., 2008; Babaeian et al., 2019). Near-surface soil moisture is also a dominant factor that regulates microbial activity and organic matter dynamics (Pries et al., 2017).

With the technological advancement of soil moisture sensors and remote sensing, the availability of soil moisture data is proliferating (e.g., Sheng et al., 2017; Babaeian et al., 2019). To gain knowledge and insight from such abundant soil moisture data, machine learning (ML) is an appealing tool as we have seen its successes in various fields, such as image

recognition, machine translation, and natural language processing. However, soil moisture data are sparsely collected with measurement errors in heterogeneous and complex soils. Therefore, it may not be enough to solely rely on a data-driven machine learning approach. An alternative approach combines machine learning with physical modeling, mainly described as differential equations. Such a hybrid approach has attracted attention in computational physics and related fields and is called scientific ML or physics-informed ML (Karniadakis et al., 2021).

Physical modeling of soil moisture is commonly conducted by solving a partial differential equation (PDE) called the Richardson Richards equation (RRE) (Richardson, 1922; Richards, 1931). Water transport properties in soils are expressed in the RRE as two relations, the water retention curve (WRC) and the hydraulic conductivity function (HCF), which are both highly non-linear functions (Assouline and Or, 2013). This double non-linearity makes it challenging to analyze and solve the RRE, which has been investigated by soil physicists and mathematicians for many decades (Philip, 1969; Radu et al., 2008;

Mitra and Vohralík, 2021). Indeed, the RRE is one of the most challenging PDEs in hydrology, and the large-scale soil moisture modeling based on the RRE has been prohibitive due to the computational demand and its unreliable solution (Paniconi and Putti, 2015; Farthing and Ogden, 2017).

Artificial neural networks (NNs) have attracted attention as an alternative numerical solver of PDEs recently. While there are some similarities between artificial NNs and biological NNs (i.e., our brain), in this context, artificial NNs should be

considered as mathematical functions with specific architectures with many adjustable parameters. Thus, artificial NNs are simply referred to as NNs in this study. Cybenko (1989) and Hornik (1991) mathematically proved that NNs can approximate any continuous function under certain conditions. The application of NNs to various fields relies on this so-called universal approximation capability to find functional relationships between available input data and target variables. Regardless of the tremendous implication of the universal approximation capability, it is not easy to find NNs that can approximate such func-

tional relationships. Therefore, significant efforts have been devoted in the machine learning community to investigate how to adjust NN parameters (or train NNs) for specific problems.

The solution to PDEs is a functional relationship between dependent and independent variables. Therefore, it is straightforward to apply NNs to approximate the solution. Lagaris et al. (1998) is among the first to use NNs to approximate the solution to boundary value problems for PDEs. To train NNs, they defined a loss function so that NNs satisfy both PDEs and

the corresponding boundary conditions and minimized the loss function to estimate the NN parameters. The minimization of the loss function required the computation of the gradient of the loss function with respect to the NN parameters as well as the partial derivatives of the PDEs. At that time, such gradients and partial derivatives were computed by manual derivation and programming. However, recent progress in the software environment (e.g., Tensorflow (Abadi et al., 2015) and Pytorch (Paszke et al., 2019)) implementing automatic differentiation (Baydin et al., 2018) enabled us to compute such gradients and partial



derivatives in a scalable manner with advanced processors such as graphics processing units (GPUs). With the technological progress, the NN-based methods to solve PDEs were reformulated as physics-informed neural networks (PINNs) by Raissi et al. (2019). PINNs have been applied to both forward and inverse modeling of PDEs in various fields, such as incompressible flows (Jin et al., 2021), subsurface transport (He et al., 2020; Tartakovsky et al., 2020), and water dynamics in soils (Bandai and Ghezzehei, 2021). Karniadakis et al. (2021) provides a comprehensive review on PINN approaches.

PINNs are particularly promising for ill-posed forward and inverse modeling. Traditional numerical solvers, such as finite difference methods (FDMs) and finite element methods (FEMs), require well-defined initial and boundary conditions, but they are often incomplete (i.e., ill-posed). This is usually the case for simulating near-surface soil moisture in real-world field condition, where obtaining accurate initial and boundary conditions is not possible without costly instrumentation (e.g., lysimeters or flux towers) (e.g., Dijkema et al., 2017).

A natural question on PINNs is whether PINNs can replace traditional numerical solvers. According to previous studies and our experiences, it is currently hard for PINNs to achieve the same accuracy as traditional methods in a competitive computational time. However, PINNs might be a competitive numerical solver for highly non-linear PDEs (e.g., the RRE and the Navier-Stokes equation) in high dimensions, where traditional numerical solvers become computationally demanding. For example, when simulating water flow under infiltration based on the RRE using traditional numerical methods, the spatial mesh
must be very small near the soil surface to obtain reliable solutions, which is computationally very expensive or intractable for a three-dimensional watershed scale. Although there is progress in numerical methods other than PINNs, including a discontinuous Galerkin method with adaptive spatial and temporal meshes (Clément et al., 2021), it is worthwhile to seek the potential of PINNs to solve the RRE.

In the study, we conducted a comprehensive analysis of PINNs as a forward and inverse numerical solver of the one-
dimensional RRE for homogeneous and heterogeneous soils. Because of the rapid progress in the field, it is impossible to test all the variants of PINNs and their training methods. Accordingly, we implemented some promising PINN methods, including layer-wise locally adaptive activation function (L-LAAF) (Jagtap et al., 2020) and domain decomposition (Jagtap and Karniadakis, 2020), where two NNs interact with each other through interface conditions to account for the discontinuity of hydraulic conductivity across layers (see Fig. 1). To validate and evaluate the solutions derived from PINNs, we compared
them to the analytical solutions given by Srivastava and Yeh (1991) and numerical solutions by an FDM and an FEM. The effects of the architecture of NNs and various parameters on the performance of PINNs were investigated. In addition to the forward modelings, we conducted inverse modeling, where a surface water flux upper boundary condition was estimated from near-surface soil moisture measurements by PINNs. Finally, we discuss current challenges and future perspectives of PINNs for forward and inverse modeling of soil moisture dynamics based on the RRE.

**Figure 1. (a)**: Schematic description of a soil consisting of two distinct layers with soil moisture sensors. **(b)**: Physics-informed neural networks (PINNs) with domain decomposition for a two-layered soil. For each input $t^{(i)}$ and $z^{(i)}$, physics and data constraints are computed through automatic differentiation. The neural network parameters $\Theta_U$ and $\Theta_L$ are estimated by minimizing the loss function $\mathcal{L}$. **(c)**: Computation by a single unit. The input values $(x_1, x_2, x_3)$ are summed with weights $(w_1, w_2, w_3)$ and added by a bias term $b$. The result is fed into the activation function $\sigma$. **(d)**: The tanh function as an adaptive activation function with a fixed scaling parameter $s$ and a trainable slope parameter $a$. The figure was inspired by Jagtap and Karniadakis (2020).





## 2 Methods

### 2.1 Richardson-Richards equation

We consider water transport in unsaturated isothermal rigid soils. In this study, hysteresis and vapor flow are ignored, and soil hydraulic properties are isotropic. The mass balance of water in a control volume implies the continuity equation:

$$\frac{\partial \theta}{\partial t} = -\nabla \cdot \mathbf{q} + S, \tag{1}$$

where $\theta$ is the volumetric water content [L$^3$ L$^{-3}$]; $t$ is the time [T]; $\mathbf{q}$ is the water flux in three dimensions [L T$^{-1}$]; $S$ is the source term [T$^{-1}$]. The water flux $\mathbf{q}$ can be derived from the Buckingham-Darcy law (Buckingham, 1907):

$$\mathbf{q} = -K(\theta)\nabla H, \tag{2}$$

where $K$ is the hydraulic conductivity [L T$^{-1}$] and $H$ is the total water head [L], which is the sum of the water potential in soils $\psi$ [L] and the elevation head $z$ (positive upward). Equations (1) and (2) are combined to derive the Richardson-Richards equation (RRE) (Richardson, 1922; Richards, 1931):

$$\frac{\partial \theta(\psi)}{\partial t} = \nabla \cdot [K(\theta)\nabla(\psi + z)] + S. \tag{3}$$

This form of the RRE is called the mixed form RRE, where both the volumetric water content $\theta$ and the water potential $\psi$ appear in the equation. To solve the RRE, the two relationships (i.e., $\theta(\psi)$ and $K(\theta)$) need to be defined. The $\theta(\psi)$ relationship is called the water retention curve (WRC), while $K(\theta)$ is referred to as the hydraulic conductivity function (HCF). The WRC and HCF are called constitutive relationships of the RRE that characterize the movement of the water in the pore space. WRCs and HCFs are commonly expressed by parametric models such as Brooks and Corey model (Brooks and Corey, 1964), van Genuchten-Mualem model (van Genuchten, 1980), and Kosugi model (Kosugi, 1996). In this study, the one-dimensional RRE without the source term $S$ is studied, which is written as

$$\frac{\partial \theta(\psi)}{\partial t} = \frac{\partial}{\partial z}\left[K(\theta)\left(\frac{\partial \psi}{\partial z} + 1\right)\right]. \tag{4}$$

The zero source term $S$ means the neglect of plant water uptake, which is valid for bare soils or soil moisture dynamics under infiltration. The one-dimensional assumption can be reasonable because water flow in near-surface soils is predominantly vertical.

### 2.2 Analytical solutions

It is difficult to obtain analytical solutions to the RRE because of the non-linearity of the WRC and the HCF. In particular, analytical solutions to the RRE for layered soils are extremely scarce. Srivastava and Yeh (1991) is one of a few analytical solutions to the transient one-dimensional RRE for both homogeneous and two-layered soils. The analytical solutions are





based on the linearization of the RRE using the following relationships for WRCs and HCFs for $\psi < 0$ (Gardner, 1958):

$$\theta = \theta_r + (\theta_s - \theta_r)e^{\alpha_G \psi}, \tag{5}$$

$$K = K_s e^{\alpha_G \psi}, \tag{6}$$

where $\theta_r$ is the residual water content [$L^3\ L^{-3}$]; $\theta_s$ is the saturated water content [$L^3\ L^{-3}$]; $\alpha_G$ is the pore-size distribution parameter [$L^{-1}$]; $K_s$ is the saturated hydraulic conductivity [$L\ T^{-1}$]. Note that the parameter $\alpha_G$ can be interpreted using van-Genuchten parameters $\alpha_{VG}$ and $n_{VG}$ (van Genuchten, 1980), as $\alpha_G \approx 1.3\alpha_{VG}n_{VG}$ (Ghezzehei et al., 2007). Although the parametric expressions for WRCs and HCFs as well as the parameter values used in the study do not necessarily represent hydraulic properties of real soils, the analytical solutions can serve to validate and assess the performance of PINN solutions
to the RRE.

### 2.2.1 Homogeneous soil

The analytical solution for a homogeneous soil requires a set of initial and boundary conditions. The lower boundary condition is a Dirichlet boundary condition $\psi = \psi_{lb}$ at $z = -Z$, where $Z$ is the vertical length of the soil [L]. The initial condition is the steady-state solution of the RRE determined by the lower boundary condition and a constant water flux upper boundary
condition $q = q_A$ at $z = 0$. The analytical solution to the time-dependent RRE with the initial and lower boundary condition as well as a constant water flux upper boundary condition $q(t) = q_B$ at $z = 0$ is written in terms of $K^* := K/K_s$:

$$K^* = q_B^* - (q_B^* - e^{\alpha_G \psi_{lb}})e^{-(z^* + Z^*)} - 4(q_B^* - q_A^*)e^{-z^*/2}e^{-t^*/4}\sum_{n=1}^{\infty}\frac{\sin(\kappa_n(z^* + Z^*))\sin(\kappa_n Z^*)e^{-\kappa_n^2 t^*}}{1 + Z^*/2 + 2\kappa_n^2 Z^*}, \tag{7}$$

where $q_A^* := q_A/K_s$; $q_B^* := q_B/K_s$; $z^* := \alpha_G z$; $Z^* := \alpha_G Z$; $t^* = \alpha_G K_s t/(\theta_s - \theta_r)$; $\kappa_n$ is the positive roots of the equation $\tan(\kappa Z^*) + 2\kappa = 0$. The analytical solution with respect to the volumetric water content $\theta$ can be computed from $K^*$ through
Eq. 5 and Eq. 6 . The explicit analytical solution clarifies larger $\alpha_G$ introduces stronger non-linearity of the solution.

### 2.2.2 Heterogeneous soil

Srivastava and Yeh (1991) provided one-dimensional analytical solutions of heterogeneous soils (i.e., two-layered soil). The analytical solution is based on the assumption that $\alpha_G$ is the same for both layers. Therefore, this analytical solution is limited to analyzing layered soils that have a discontinuity in the hydraulic conductivity $K$ across the layers. In fact, the volumetric
water content $\theta$ is continuous across the layers as the water potential $\psi$. The initial and boundary conditions are the same as the homogeneous case. The analytical solution is much more complicated than the homogeneous one, so we refer to the original literature for the detail (Srivastava and Yeh, 1991). However, we provide the computed analytical solutions and numerical derivations on Bandai and Ghezzehei (2022), which can be useful to validate other numerical methods.





## 2.3 Mathematical formulation of PINNs

### 2.3.1 Feedforward neural networks

Feedforward NNs are used to approximate the solution of PDEs in PINNs. In this section, the mathematical formulation of feedforward NNs with $L$ hidden layers with layer-wise locally adaptive activation functions (L-LAAFs) (Jagtap et al., 2020) is introduced. NNs are mathematical functions $\mathcal{N}$ mapping an input vector $\mathbf{x} \in \mathbb{R}^{n^x}$ to an output vector $\hat{\mathbf{y}} \in \mathbb{R}^{n^y}$:

$$\hat{\mathbf{y}} := \mathcal{N}(\mathbf{x}). \tag{8}$$

The hat operator represents prediction throughout the paper. NNs are often represented as layers of units (or neurons), as in Fig. 1 (b), where two feedforward NNs consisting of four hidden layers with six units are shown.

NNs are compositions of affine transformations (the composition of linear tranformation and translation) and non-linear functions. Herein, $\mathbf{h}^{[k]} \in \mathbb{R}^{n^{[k]}}$ for an integer $k$ such that $1 \leq k \leq L$ represents the vector value corresponding to the $k$th hidden layer consisting of $n^{[k]}$ units. $\mathbf{h}^{[k]}$ for each $k$ is computed in the following manner:

$$
\begin{aligned}
\mathbf{h}^{[1]} &:= \sigma(sa^{[1]}(\mathbf{W}^{[1]}\mathbf{x} + \mathbf{b}^{[1]})), \\
\mathbf{h}^{[2]} &:= \sigma(sa^{[2]}(\mathbf{W}^{[2]}\mathbf{h}^{[1]} + \mathbf{b}^{[2]})), \\
&\vdots \\
\mathbf{h}^{[L-1]} &:= \sigma(sa^{[L-1]}(\mathbf{W}^{[L-1]}\mathbf{h}^{[L-2]} + \mathbf{b}^{[L-1]})), \\
\mathbf{h}^{[L]} &:= \sigma(sa^{[L]}(\mathbf{W}^{[L]}\mathbf{h}^{[L-1]} + \mathbf{b}^{[L]})),
\end{aligned}
\tag{9}
$$

where $\mathbf{W}^{[k]}$ and $\mathbf{b}^{[k]}$ are the weight matrix and bias vector for the $k$th hidden layer; $s \geq 0$ is a fixed scaling factor; $a^{[k]}$ represents a trainable parameter changing the shape of the element-wise activation function $\sigma$. The output of the NN is computed as

$$\hat{\mathbf{y}} := o(\mathbf{W}^{[L+1]}\mathbf{h}^{[L+1]} + \mathbf{b}^{[L+1]}), \tag{10}$$

where $o$ is the output function; $\mathbf{W}^{[L+1]}$ and $\mathbf{b}^{[L+1]}$ are the weight matrix and bias vector for the output layer. The collection of the weight matrices $\mathbf{W} := \{\mathbf{W}^{[1]}, ..., \mathbf{W}^{[L+1]}\}$, the bias vectors $\mathbf{b} := \{\mathbf{b}^{[1]}, ..., \mathbf{b}^{[L+1]}\}$, and the slope parameters
$\mathbf{a} := \{a^{[1]}, ..., a^{[L]}\}$ are the parameters of the NN, which are denoted by $\Theta := \{\mathbf{W}, \mathbf{b}, \mathbf{a}\}$ in this paper.

To understand the role of the parameters $s$ and $a^{[k]}$ introduced in the L-LAAF (Jagtap et al., 2020), consider a case where $sa^{[k]} = 1$ for all $k$, and $\sigma$ is the identity function. In this case, the neural network $\mathcal{N}$ is nothing but an affine transformation and cannot learn a non-linear relationship. In a standard NN, non-linear activation functions, such as the hyperbolic tangent function (tanh), are used with $sa^{[k]} = 1$ for all $k$ to learn non-linear relationships between input and output variables. As shown
in Fig. 1 (**d**), the tanh function has a "linear" regime near the origin and exhibits the non-linearity outside the region. By increasing the parameter $sa^{[k]}$, we can increase the slope of the activation function and narrow the "linear" regime. Jagtap et al. (2020) reported that larger $sa^{[k]}$ accelerated the training of PINNs and captured the high-frequency components of the solution of PDEs, while too large $sa^{[k]}$ made the training unstable. Throughout the study, $a^{[k]}$ was initialized to 0.05, and $s$ was set to 20 when the L-LAAF was used.





### 2.3.2 Formulation of PINNs for the RRE

In this section, PINNs to solve the forward and inverse problems for the RRE are described. First, we consider the forward modeling of the one-dimensional RRE defined on a domain $\Omega = (-Z, 0)$, the boundary $\partial\Omega$, and the time [0, T]:

$$\frac{\partial \theta}{\partial t} = \frac{\partial}{\partial z}\left[ K \left( \frac{\partial \psi}{\partial z} + 1 \right) \right], \quad z \in \Omega, \quad t \in (0, T), \tag{11}$$

$$\theta(z, 0) = g(z), \quad z \in (\Omega \cup \partial\Omega), \tag{12}$$

$$\theta(z, t) = h(z, t), \quad z \in \partial\Omega_D, \quad t \in (0, T), \tag{13}$$

$$q(z, t) := -K(z, t)\left( \frac{\partial \psi(z, t)}{\partial z} + 1 \right) = i(z, t), \quad z \in \partial\Omega_F, \quad t \in (0, T), \tag{14}$$

where $g(z)$ is the initial condition; $h(z, t)$ is the Dirichlet boundary condition on the Dirichlet boundary $\partial\Omega_D$; $q(z, t)$ is the water flux in the vertical direction (positive upward); $i(z, t)$ is the water flux boundary condition on the flux boundary $\partial\Omega_F$. Here, we only use the volumetric water content $\theta$ for the initial condition and the Dirichlet boundary condition because the measurement of $\theta$ is more reliable than the water potential $\psi$ in practical situations, though the modification to the water potential $\psi$ is straightforward. Although we limit ourselves to either the Dirichlet or the water flux boundary condition, the framework can be extended to other boundary conditions. In this study, we focus on a particular situation where the soil surface ($z = 0$) is set to $\partial\Omega_F$, and the bottom ($z = -Z$) is set to $\partial\Omega_D$, which corresponds to when soil moisture dynamics is controlled by the surface water flux $q(0, t)$ (i.e., evaporation or infiltration) and the volumetric water content at the bottom $\theta(-Z, t)$.

PINNs aim to approximate the solution of the one-dimensional RRE $\psi(z, t)$ by a NN $\mathcal{N}(z, t)$ with the NN parameters $\Theta = \{\mathbf{W}, \mathbf{b}, \mathbf{a}\}$. Because the water potential $\psi$ is negative in unsaturated soils, we used the identity function for the output function $o$ of the NN (Eq. 10) and transformed the output as

$$\hat{\psi}(z, t) := -\exp(\mathcal{N}(z, t; \Theta)) + \beta, \tag{15}$$

where $\beta$ is a fixed parameter, which can allow $\hat{\psi}(z, t)$ to be zero or positive (saturated) and gives PINNs better initial guess of the solution for certain problems. To construct PINNs for the RRE, the residual of the RRE is defined as:

$$\hat{r}(z, t; \Theta) := \frac{\partial \hat{\theta}}{\partial t} - \frac{\partial}{\partial z}\left[ \hat{K} \left( \frac{\partial \hat{\psi}}{\partial z} + 1 \right) \right]. \tag{16}$$

Here, $\hat{\theta}$ and $\hat{K}$ are computed from $\hat{\psi}$ with the predefined WRC and HCF (i.e., $\theta(\psi)$ and $K(\theta)$). The partial derivatives in the residual are computed through the reverse mode automatic differentiation (Baydin et al., 2018). The collection of the NN parameters $\Theta$ are identified by minimizing a loss function, which is defined as

$$\mathcal{L}(\Theta) := \lambda_r \mathcal{L}_r(\Theta) + \sum_i \lambda_i \mathcal{L}_i(\Theta), \tag{17}$$

where $\lambda_r$ is the weight parameter corresponding to the loss term for the residual of the RRE $\mathcal{L}_r(\Theta)$; $\lambda_i$ is the weight parameter for the loss term $\mathcal{L}_i(\Theta)$ for $i = \{m, ic, D, F\}$, where $m, ic, D$, and $F$ represent the measurement data, the initial condition, and





the Dirichlet and the water flux boundary conditions, respectively. $\mathcal{L}_r(\Theta)$ and $\mathcal{L}_i(\Theta)$ for $i = \{m, ic, D, F\}$ are defined as:

$$\mathcal{L}_r(\Theta) := \frac{1}{N_r} \sum_{i=1}^{N_r} [\hat{r}(z_r^i, t_r^i)]^2, \tag{18}$$


$$\mathcal{L}_m(\Theta) := \frac{1}{N_m} \sum_{i=1}^{N_m} [\hat{\theta}(z_m^i, t_m^i) - \theta_m^i]^2, \tag{19}$$

$$\mathcal{L}_{ic}(\Theta) := \frac{1}{N_{ic}} \sum_{i=1}^{N_{ic}} [\hat{\theta}(z_{ic}^i, 0) - g(z_{ic}^i)]^2, \tag{20}$$

$$\mathcal{L}_D(\Theta) := \frac{1}{N_D} \sum_{i=1}^{N_D} [\hat{\theta}(z_D^i, t_D^i) - h(z_D^i, t_D^i)]^2, \tag{21}$$

$$\mathcal{L}_F(\Theta) := \frac{1}{N_F} \sum_{i=1}^{N_F} [\hat{q}(z_F^i, t_F^i) - i(z_F^i, t_F^i)]^2, \tag{22}$$

where $\{z_r^i, t_r^i\}_{i=1}^{N_r}$ denotes the $N_r$ residual points (also called collocation points) at which the residual of the PDE is evaluated; $\{\theta_m^i, z_m^i, t_m^i\}_{i=1}^{N_m}$ denotes the $N_m$ measurement data points; $\{z_{ic}^i\}_{i=1}^{N_{ic}}$ denotes the $N_{ic}$ initial condition points; $\{z_D^i, t_D^i\}_{i=1}^{N_D}$
denotes the $N_D$ Dirichlet boundary condition points; $\{z_F^i, t_F^i\}_{i=1}^{N_F}$ denotes the $N_F$ water flux boundary condition points. Here, $\mathcal{L}_r$ forces PINNs to satisfy the PDE, and $\mathcal{L}_m$ helps PINNs to fit the measurement data while $\mathcal{L}_{ic}$, $\mathcal{L}_D$, and $\mathcal{L}_F$ enforce the initial condition, the Dirichlet and the water flux boundary conditions, respectively. Note that initial and boundary conditions can be encoded in a hard manner so that the approximated solution automatically satisfies these conditions (Lagaris et al., 1998; Sun et al., 2020). However, we encode these conditions in a soft manner and treat them as data points as in Eq. 17 to leverage the
flexibility of PINNs, which is essential in practical situations where precise initial and boundary conditions are rarely available.

In the framework, the measurement data can be provided, which is not necessary to make the forward modeling well-posed. PINNs can easily incorporate such additional measurement data to improve accuracy and computational efficiency. As for the inverse modeling, the implementation of the PINNs is almost identical to the forward modeling. If we invert physical parameters in PDEs from measurement data (e.g., saturated hydraulic conductivity $K_s$), the parameters and the NN parameters
are simultaneously estimated. Also, one can drop the loss terms for the initial or boundary conditions if they are not available, though this makes the problem ill-posed. In the study, the forward and the inverse modeling was conducted in Sect. 3 and Sect. 4, respectively.

### 2.3.3 Errors in PINN solutions

Despite the increasing popularity and successes of PINNs in various fields, the theoretical understanding of PINNs is still lim-
ited. Shin et al. (2020) is among the first who conducted a rigorous analysis of PINNs, where they formulated the generalization





error of PINNs as the sum of the approximation error, the optimization error, and the estimation error. The approximation error is the distance between the best possible approximation by PINNs and the solution of PDEs. The optimization error is due to the difficulty in minimizing the non-linear and non-convex loss function. The estimation error is caused by the insufficiency of data to train PINNs. They demonstrated theoretically and numerically that the sum of the approximation and the estimation
error decreased with the increase in the training data for linear second-order elliptic and parabolic PDEs.

Mishra and Molinaro (2022) provided a theoretical framework to estimate the generalization error of PINNs for a variety of PDEs, including non-linear PDEs. They demonstrated that the generalization error would be sufficiently low if 1) PINNs are trained well (i.e., small optimization error); 2) the number of residual points is large; 3) the solution of PDEs is sufficiently regular. In the study, we numerically analyzed the accuracy of PINN solutions to the RRE and investigated whether their
theoretical claims could be applied to our case.

## 2.4 Implementation of PINNs

Training of NNs requires trial and error because the theoretical understanding of the mechanism is still limited. However, feedforward NNs used in PINNs have been investigated for many years, so empirical knowledge is available (Bengio, 2012; LeCun et al., 2012). Those techniques are not new, but we would like to reiterate some of them with the explanation of our
implementation of PINNs. The PINN algorithm for heterogeneous soils is summarized in Fig. 2. We used TensorFlow (Abadi et al., 2015) to implement PINNs, and the source code is available on Bandai and Ghezzehei (2022).

### 2.4.1 Architecture of neural networks

Before training NNs, it is recommended to transform input data so that the components of input variables $\mathbf{x}$ have zero mean, and each variable has a similar variance. In our implementation, we did not transform or normalize input data (i.e., $t \in [0, T]$
and $z \in [-Z, 0]$). However, when both input variables were positive, the training of PINNs was difficult, as mentioned in LeCun et al. (2012). Thus, it is better to transform input data for future studies. As for output variables, it is also important to take into account their range, as in Eq. 15.

The architecture of NNs (i.e., the number of hidden layers $L$ and units $n^{[k]}$ for $k = 1, ..., L$) determines the complexity of functions the NN can learn and thus depends on PDEs of interest and the corresponding initial and boundary conditions. It is
known that the expressive capability of NNs grows exponentially with the number of hidden layers (Raghu et al., 2017). We used the same number of units for all hidden layers, and the effects of the architecture of NNs on PINN performance were experimentally investigated. As for activation functions, symmetric activation functions with respect to the origin, such as the tanh function, are preferable to non-symmetric functions such as the sigmoid function. Because PINNs require the second derivative of state variables in the loss function, we used the tanh function for the activation functions $\sigma$ for all hidden layers.





---

**Algorithm 1** PINNs with domain decomposition

---

**Step 0**: Divide the spatial domain into $\Omega_U$ and $\Omega_L$ and assign two neural networks $\mathcal{N}_U$ and $\mathcal{N}_L$ to each domain. Determine the architecture of two neural networks $\mathcal{N}_U$ and $\mathcal{N}_L$ and weight parameters $\lambda_i$ in the loss function $\mathcal{L}$ (Eq. 17).

**Step 1**: Construct the neural networks $\mathcal{N}_U(z, t; \Theta_U)$ and $\mathcal{N}_L(z, t; \Theta_L)$ with neural network parameters $\Theta_U$ and $\Theta_L$.

**Step 2**: Initialize the neural network parameters to $\Theta_U^0$ and $\Theta_L^0$.

**Step 3**: Given available data (e.g., initial and boundary conditions, measurement data), train the neural networks $\mathcal{N}_U$ and $\mathcal{N}_L$ by minimizing the loss function $\mathcal{L}(\Theta_U, \Theta_L)$.

i ← 0

**while** $i <$ max_iteration_Adam **do**
    $\Theta_U^{i+1} \leftarrow \Theta_U^i + \hat{\Theta}_U^i$
    $\Theta_L^{i+1} \leftarrow \Theta_L^i + \hat{\Theta}_L^i$
    i ← i + 1
**end while**

**while** L-BFGS-B stopping criteria are not met **do**
    $\Theta_U^{i+1} \leftarrow \Theta_U^i + \hat{\Theta}_U^i$
    $\Theta_L^{i+1} \leftarrow \Theta_L^i + \hat{\Theta}_L^i$
    i ← i + 1
**end while**

---

**Figure 2.** The algorithm for PINNs with domain decomposition for a two-layered soil. $\Omega_U$ and $\Omega_L$ refer to the spatial domain for the upper and lower layers, respectively. In Step 3, max_iteration_Adam is the maximum number of the Adam iteration (set to 100000 in the study); $\hat{\Theta}$ represents the update for the neural network parameter $\Theta$; L-BFGS-B stopping criteria are summarized in Sect. 2.4.3.

### 2.4.2 Initialization

At the beginning of the training, the collection of the weight parameters $\mathbf{W}$ and the bias parameters $\mathbf{b}$ have to be initialized. Glorot and Bengio (2010) demonstrated that simple random initialization caused the activation functions to be "saturated," meaning that the slope of the activation function becomes zero. To prevent this, they introduced the Xavier initialization for the weight parameters $\mathbf{W}$, which was used in our implementation. The bias parameters $\mathbf{b}$ were initialized to be 0. Because the initialization of $\mathbf{W}$ significantly affects PINN solutions, different sets of randomization must be tested. Therefore, we used ten different random seeds for the initialization of each setting of PINNs.

### 2.4.3 Training

PINNs were trained to minimize the loss function (Eq. 17). The loss term for the residual $\mathcal{L}_r$ was evaluated at randomly sampled residual points in the spatial and temporal domain (Fig. S1 (**b**)). For each problem, the same residual points were used. We





tested the residual-based adaptive refinement algorithm proposed by Lu et al. (2021b), where residual points are chosen where the residual of PDEs is high. For our case study, the effectiveness of the algorithm was minor, and thus the results are only shown in the supplementary material (see Sect. S1.1 and Fig. S1).

The weight parameters in the loss function $\lambda_i$ (Eq. 17) play a crucial role in minimizing the loss function. In the original PINN framework proposed by Raissi et al. (2019), all $\lambda_i$ were set to 1. However, Wang et al. (2021) demonstrated that the
loss terms corresponding to initial and boundary conditions need to be penalized more, and optimal values of those weight parameters are problem-dependent. To overcome this challenge, they proposed the learning rate annealing algorithm (Wang et al., 2021), where the weight parameters in the loss function $\lambda_i$ are updated during training to balance the relative importance of each loss term. We tested the algorithm but resulted in a modest improvement compared to the L-LAAF, so the results are given in the supplementary material (Sect. S1.2 and Fig. S2). In the study, the effects of $\lambda_i$ were investigated in Sect 3.1.5.

It is common to use a stochastic gradient descent algorithm to minimize the loss function to train NNs. In this study, we used the Adam algorithm (Kingma and Ba, 2014). Because the Adam optimizer is not enough to achieve solutions with high accuracy, previous studies on PINNs employed a two-step optimization strategy, and we followed it. First, the loss function was minimized using $10^5$ iterations of the Adam algorithm in TensorFlow (Abadi et al., 2015) with the exponential decay of the learning rate. The initial learning rate was set to 0.001 with the decay rate of 0.90, the decay step was set to 1000, and
the other parameters were set to their default values. The Adam algorithm used a "mini-batch" of the data, where only 128 of all residual points were considered while all the initial and boundary data points were used for each iteration. After the Adam algorithm, the loss function was further minimized through the L-BFGS-B optimizer (Byrd et al., 1995) from Scipy (Virtanen et al., 2020), which was terminated once the loss function converged with prescribed thresholds. The L-BFGS-B algorithm can utilize the information on the approximated curvature of the loss function and find a better local minimum than a stochastic
gradient descent for the case of PINNs. The following L-BFGS-B parameters were used: maxcor = 50, maxls = 50, maxiter = 50000, maxfun = 50000, ftol = $1.0 \times 10^{-10}$, gtol = $1.0 \times 10^{-8}$, and the default values for the other parameters. Although it is desirable to tune the parameters for each optimizer, we fixed those parameters in the study.

### 2.4.4  Domain decomposition

Natural soils have distinctive layering, and the hydraulic properties of each layer vary between the layers, which results in
continuous but not a differentiable water potential distribution in the soil profile. To deal with such spatial heterogeneity, Jagtap and Karniadakis (2020) proposed a domain decomposition method for PINNs, where a computational domain is divided into sub-domains, and a NN is assigned to each sub-domain. Then, the NNs interact with each other during the training through interface conditions such as the continuity of mass and flux. Such interface conditions can be incorporated into the loss function. For simulating water flow in a two-layered soil, two NNs $\mathcal{N}_U$ and $\mathcal{N}_L$ are assigned to the upper and lower layer, and the





continuities of water potential, water flux, and the residual of the RRE are imposed at the boundary:

$$\hat{\psi}_U(z_I, t) = \hat{\psi}_L(z_I, t), \tag{23}$$

$$\hat{q}_U(z_I, t) = \hat{q}_L(z_I, t), \tag{24}$$

$$\hat{r}_U(z_I, t) = \hat{r}_L(z_I, t), \tag{25}$$

where the subscripts $_U$ and $_L$ mean a value with the subsucripts (e.g., $\hat{\psi}_U$) was computed by $\mathcal{N}_U$ and $\mathcal{N}_L$, respectively; $z_I$ represents the spatial coordinate of the interface. These interface conditions are incorporated into the loss function as a loss term (Eq. 17):

$$\mathcal{L}_{I_\psi}(\Theta_U, \Theta_L) := \frac{1}{N_I} \sum_{i=1}^{N_I} [\hat{\psi}_U(z_I, t^i) - \hat{\psi}_L(z_I, t^i)]^2, \tag{26}$$

$$\mathcal{L}_{I_q}(\Theta_U, \Theta_L) := \frac{1}{N_I} \sum_{i=1}^{N_I} [\hat{q}_U(z_I, t^i) - \hat{q}_L(z_I, t^i)]^2, \tag{27}$$

$$\mathcal{L}_{I_r}(\Theta_U, \Theta_L) := \frac{1}{N_I} \sum_{i=1}^{N_I} [\hat{r}_U(z_I, t^i) - \hat{r}_L(z_I, t^i)]^2, \tag{28}$$

where $N_I$ is the number of points on the interface, where the loss terms are evaluated; $\Theta_U$ and $\Theta_L$ are the neural network parameters for $\mathcal{N}_U$ and $\mathcal{N}_L$, respectively. In the implementation, we found that the logarithmic transformation of water potential $\psi$ was helpful to balance the loss terms. Therefore, instead of $\mathcal{L}_{I_\psi}$, we imposed the continuity of the output of the neural networks, as in

$$\mathcal{L}_{I_\mathcal{N}}(\Theta_U, \Theta_L) := \frac{1}{N_I} \sum_{i=1}^{N_I} [\mathcal{N}_U(z_I, t^i) - \mathcal{N}_L(z_I, t^i)]^2. \tag{29}$$

Note that the original literature Jagtap and Karniadakis (2020) trained each NN separately by constructing multiple loss functions for parallel computation, but we trained the two NNs ($\mathcal{N}_U$ and $\mathcal{N}_L$) simultaneously. The algorithm is summarized in Fig. 2. Although our algorithm is for a two-layered soil in one dimension, this method can be extended to more layers with more complex geometries in higher dimensions (Jagtap and Karniadakis, 2020).

### 2.5 Evaluation of numerical error

Numerical errors of PINNs and the other numerical methods (i.e., FDMs and FEMs) in terms of the volumetric water content $\theta$ and the water potential $\psi$ were evaluated by comparing the analytical solutions to those numerical solutions computed on a uniform grid with a spatial step of 0.1 cm and temporal step of 0.1 h. Absolute error, relative absolute error, squared error, and relative squared error were computed. Because those different types of errors exhibited strong correlations, we show, in the following sections, relative squared error in terms of the volumetric water content $\epsilon^\theta$ defined as:

$$\epsilon^\theta := \frac{\sum_t \sum_z (\hat{\theta}(t, z) - \theta(t, z))^2}{\sum_t \sum_z \theta(t, z)^2}, \tag{30}$$

where $\theta$ and $\hat{\theta}$ represent analytical and numerical solutions, respectively.





## 3 Forward modeling

In this section, we report the main results of the forward modeling of water transport in homogeneous and heterogeneous soils using PINNs. PINN solutions of the RRE with varying NN architectures and parameters were evaluated using the one-

dimensional analytical solutions by Srivastava and Yeh (1991) for homogeneous and heterogeneous soils introduced in Sect. 2.2. To assess the performance of PINNs, numerical solutions obtained by an FDM and an FEM are also presented. The implementation of the FDM for the homogeneous case is described in Sect. S1.3 while the FEM solution for the heterogeneous case was obtained by HYDRUS-1D (Šimůnek et al., 2013).

### 3.1 Homogeneous soil

We simulated water infiltration into a homogeneous soil using the RRE. This simple setup was used to understand the characteristics of PINNs with different settings. We specifically investigated 1) the effects of NN architecture (the number of layers and units as well as the use of the layer-wise locally adaptive activation function (L-LAAF)); 2) the effects of the weight parameters $\lambda_i$ in the loss function (Eq. 17); 3) the effects of the number of the residual points and the upper boundary data points.

### 3.1.1 Problem setup

We considered soil moisture dynamics in a homogeneous soil induced by a constant water flux on the surface ($z = 0$ cm), as introduced in Sect. 2.2.1, where $Z = 10$ cm, $T = 10$ h, $q_A = -0.1$ cm h$^{-1}$, $q_B = -0.9$ cm h$^{-1}$, $\psi_{lb} = 0$ cm, $\theta_r = 0.06$; $\theta_s = 0.40$; $\alpha_G = 1.0$ cm$^{-1}$; $K_s = 1.0$ cm h$^{-1}$.

### 3.1.2 Characteristics of PINN solution

The numerical solution by PINNs and the analytical solution are shown in Fig. 3 (**a**). The PINN solution was obtained by using a NN of 5 hidden layers with 50 units using the L-LAAF, which was determined after testing various settings described in the following sections. $N_{ic} = 101$ initial data points ($z = -10.0, -9.9, ..., -0.1, 0.0$ cm), $N_{ub} = 1000$ upper boundary data points ($t = 0.01, 0.02, ..., 9.99, 10.0$ h), and $N_{lb} = 100$ lower boundary data points ($t = 0.1, 0.2, ..., 9.9, 10.0$ h) were used to train the NN. The number of residual points $N_r$ was set to 10000. The weight parameters $\lambda_i$ in the loss function were set to ten for

the initial condition ($\lambda_{ic}$), the lower boundary condition ($\lambda_{lb}$), and the water flux upper boundary condition ($\lambda_{ub}$) while $\lambda_r$ was set to one for the residual loss term. The difference between the PINN and the analytical solution in the volumetric water content $\theta$ is shown in the right column of Fig. 3 (**a**). Larger errors were observed near the initial and upper boundary conditions, where strong non-linearity exists due to the surface water flux. Except for this, PINNs could approximate the solution with high accuracy. Figure 3 (**b**) showed the FDM solution with a spatial mesh $dz = 0.1$ cm and a time step $dt = 0.01$ h, which

is comparable to the temporal resolution of the upper boundary data points given to the PINN. In comparison with the FDM solution, the PINN solution was quite reasonable ($\epsilon^\theta = 4.86 \times 10^{-4}$ for the PINN and $\epsilon^\theta = 9.72 \times 10^{-4}$ for the FDM solutions, respectively). Note that the degree of freedom of the FDM solution was 101000, while the number of parameters of the NN





**(a)** Physics-informed neural networks

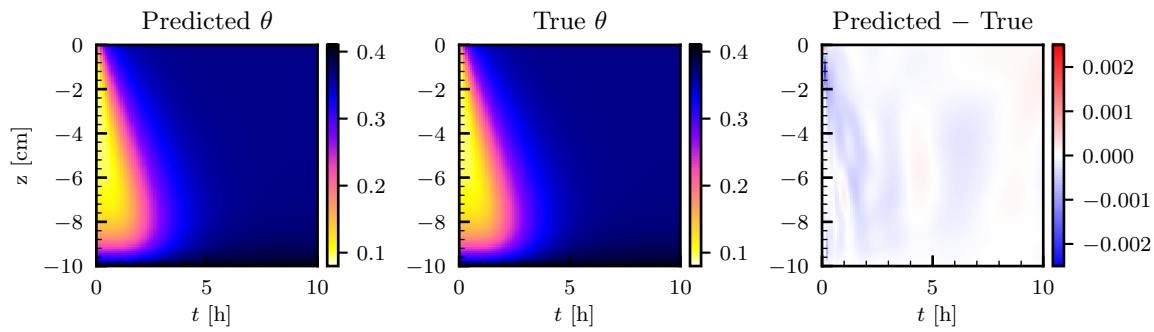

**(b)** Fintie difference method

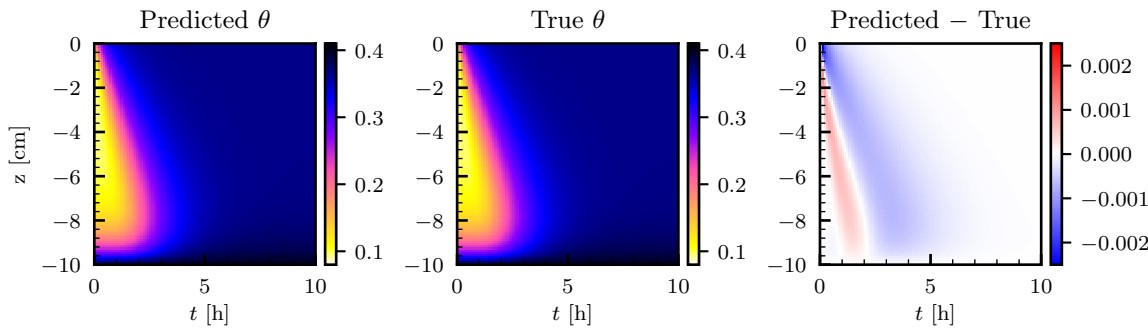

**Figure 3.** Homogeneous soil. (**a**): Physics informed neural network (PINN) solution in terms of volumetric water content $\theta$ [-] (left column). The neural network consisted of 5 hidden layers with 50 units with the layer-wise adaptive activation function. True analytical solution (center column) is given by Srivastava and Yeh (1991) (see Sect. 2.2.1), and the difference between the PINN and true solutions are shown in the right column. (**b**): Numerical solution by a finite difference method with a spatial mesh of $dz = 0.1$ cm and a time step $dt = 0.01$ h.

was 10406, which demonstrates the memory efficiency of PINNs. Also, the number of residual points of PINNs was much smaller than the degree of freedom of the FDM. However, increasing the number of residual points did not improve the PINN

solution, as discussed in Sect. 3.1.6, while the FDM solution improved by further minimizing $dz$ and $dt$ ($\epsilon^{\theta} = 1.03 \times 10^{-5}$ was obtained for $dz = 0.01$ cm and $dt = 0.0001$ h, as shown in Fig. S3). It is important to note here that an FDM with such a very fine mesh size requires solving a large linear system multiple times for each time step because the RRE is a non-linear PDE, and the number of the iteration increases with decreasing $dz$, which leads to significant computational demand. This situation becomes worse for higher dimensions. Although we do not test PINNs for higher dimensions, other studies demonstrated the

effectiveness of PINNs for higher dimensions (Mishra and Molinaro, 2022). This is the main reason we see the potential of PINNs for a large-scale simulation based on the RRE.





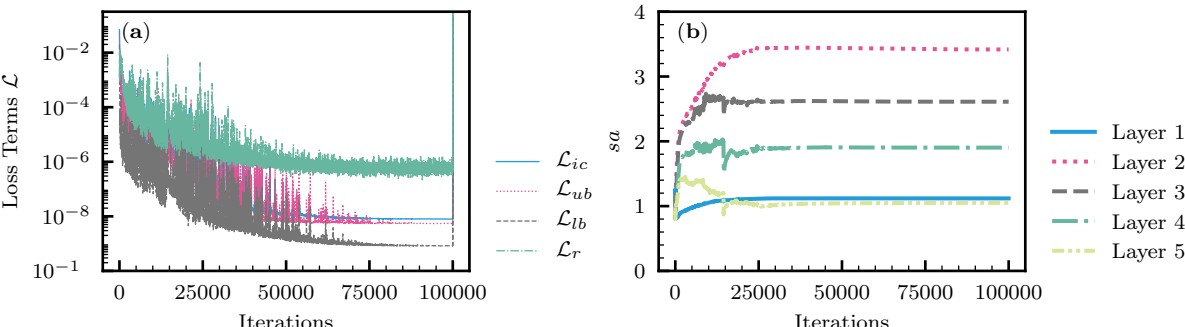

**Figure 4.** Homogeneous soil. (**a**): The evolution of the loss terms for the initial condition $\mathcal{L}_{ic}$, the upper boundary condition $\mathcal{L}_{ub}$, the lower boundary condition $\mathcal{L}_{lb}$, and the residual $\mathcal{L}_r$ during the Adam (100000 iterations) and the following L-BFGS-B training. (**b**): The evolution of the adaptive parameter $sa$ for layer-wise locally adaptive activation functions (Eq. 9) for each hidden layer (Layer 1 is next to the input layer).

### 3.1.3 Training PINNs

A typical evolution of the loss terms in the loss function during the training (the Adam and L-BFGS-B algorithms) is shown in Fig. 4 (**a**). In most cases, the Adam algorithm gave a good minimum of the loss function, and the following L-BFGS-B
algorithm met its termination criteria immediately (a spike was observed just before the termination). Among the loss terms, the residual loss $\mathcal{L}_r$ remained high after the training ($\mathcal{L}_r \approx 10^{-6}$). $\mathcal{L}_r$ indicates whether the RRE was satisfied in the spatial and temporal domain and determines the performance of PINNs. The characteristics of $\mathcal{L}_r$ is further explored in Sect. 3.1.7. Figure 4 (**b**) shows the evolution of the adaptive parameter $sa$ for the L-LAAF for each hidden layer (Eq. 9). The parameter $sa$ changes the slope and the linear regime of activation functions, as shown in Fig. 1 (**d**). The parameter $sa$ varied with the
iterations of the Adam algorithm and reached its limiting value for each hidden layer. $sa$ for the second hidden layer was the highest, and smaller $sa$ was observed for hidden layers closer to the output layer.

Figure 5 demonstrates how PINNs learn the solution to the RRE during the training. At the initialization (Fig. 5 (**a**)), the PINN solution greatly differed from the true solution. However, PINNs quickly learned the lower boundary condition (see Fig. 5 (**b**)). Although the initial condition was given as data points, it took more iterations for PINNs to learn it because of the high
non-linearity near the surface induced by the water flux boundary condition. This corresponds to the increase in the adaptive parameter $sa$ for the L-LAAF (see Fig. 4 (**b**)). The limiting value of $sa$ for the second hidden layer was 3.4, which makes the tanh function highly non-linear and closer to the step function (see Fig. 1 (**d**)). We concluded that the L-LAAF helped PINNs learn the highly non-linear solution of the RRE. Figure 5 clearly illustrated that PINNs first learned less complex parts of the solution and then captured the more complex parts. The tendency of feedforward NNs to learn less complex functions is called
"spectral bias," and Wang et al. (2022) demonstrated that this spectral bias caused PINNs to fail to learn complex solutions of PDEs. Further research is needed on how PINNs can learn more complex solutions of the RRE, for example, where wetting and drying cycles are studied.

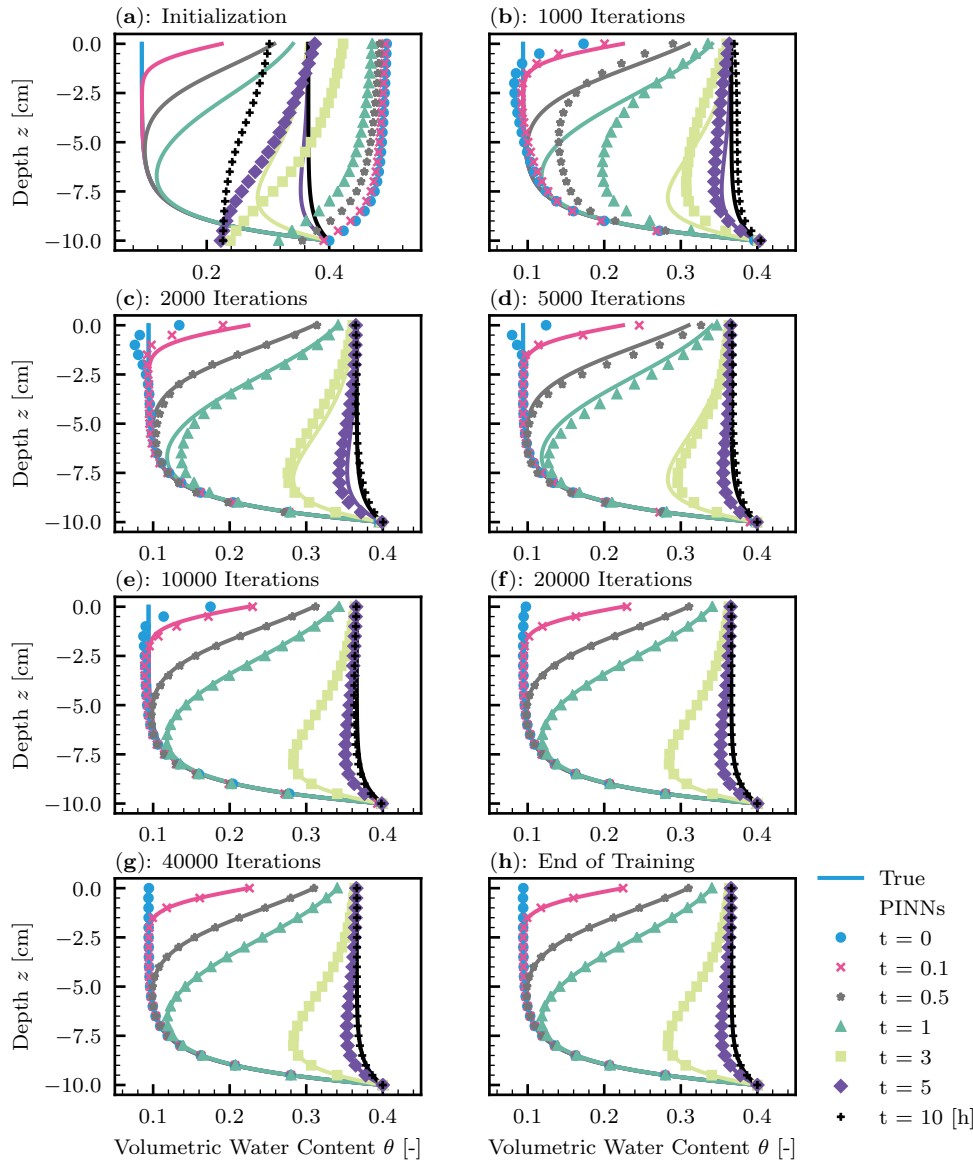

**Figure 5.** Homogeneous soil. The evolution of physics-informed neural network (PINN) solution (points) during the training with the true solution (solid lines). (**a**): Initialization of PINNs. (**b**) to (**g**): 1000, 2000, 5000, 10000, 20000, 40000 iterations of the Adam algorithm. (**h**): The end of 100000 iterations of the Adam algorithm and the following L-BFGS-B algorithm.





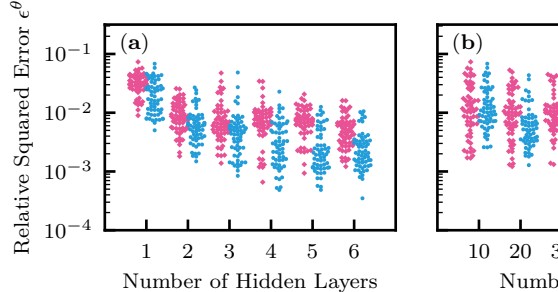 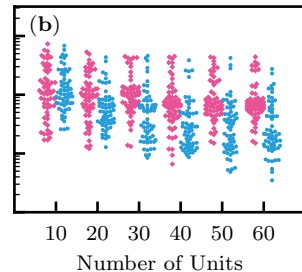 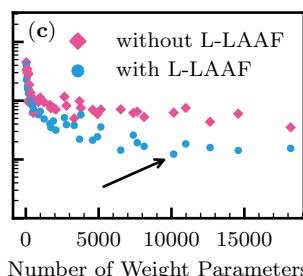

**Figure 6.** Homogeneous soil. Relative squared error in terms of volumetric water content $\epsilon^\theta$ for different numbers of hidden layers (**a**) and units (**b**) with and without the use of the layer-wise locally adaptive activation function (L-LAAF). In Fig. (**c**), the averaged $\epsilon^\theta$ were computed for each neural network architecture, and the values were plotted against the number of the weight parameters of each neural network. The arrow indicates the lowest averaged $\epsilon^\theta$, which corresponds to neural networks with 5 hidden layers with 50 units used in Fig. 3

### 3.1.4 Effects of neural network architecture

The effects of the number of hidden layers and units of NNs as well as the use of the L-LAAF were investigated. The candidate numbers of hidden layers and units were 1, 2, 3, 4, 5, 6 for hidden layers and 10, 20, 30, 40, 50, 60 for units. The L-LAAF was turned on and off, and ten different random seeds were used for the NN initialization for each NN architecture, which resulted in a total of 720 runs.

The summary of the results is shown in Fig. 6. In Fig. 6 (**a**) and (**b**), all the 720 data points are plotted while the averaged data for each NN architecture are shown in Fig. 6 (**c**) against the number of the weight parameters of each NN architecture. In Fig. 6 (**a**), smaller relative squared error $\epsilon^\theta$ were observed for a larger number of hidden layers. Four hidden layers appeared to be enough for NNs to approximate the solution. As for the number of units, increasing the number gave smaller $\epsilon^\theta$ though the effect seemed less relevant than that for hidden layers. It is clear from Fig. 6 (**c**) that the use of the L-LAAF improved the performance of PINNs. From this analysis, we determined the best NN architecture to be 5 hidden layers with 50 units with the L-LAAF indicated by the arrow in Fig. 6 (**c**), whose solution is shown in Fig. 3.

### 3.1.5 Effects of weight parameters in loss function

The effects of the weight parameters $\lambda_i$ in the loss function (Eq. 17) were studied by varying $\lambda_i$ for the initial and boundary conditions while $\lambda_r$ was fixed to one. We denote $\lambda_{ub}$ and $\lambda_{lb}$ as the weight parameters for the upper and lower boundary conditions, respectively. Five different values (1, 3, 10, 30, 100) were tested for each weight parameter with ten different NN initializations, which resulted in a total of 1250 simulations, where the NN architecture was fixed to 5 hidden layers with 50 units with the L-LAAF.

Figure 7 shows the effects of the weight parameters $\lambda_i$ for $i = ic, ub, lb$ on the loss terms $\mathcal{L}_i$ in the loss function and the PINN performance evaluated by the relative squared error $\epsilon^\theta$. Each panel in the figure contains all the 1250 simulations, and



the effects of the initialization and the $\lambda_i$ are mixed. Figure 7 (**a**), (**e**), and (**i**) demonstrated that higher weight parameters $\lambda_i$ attained lower values of the corresponding loss terms $\mathcal{L}_i$, which was expected. Another noticeable feature is that the loss term

for the residual $\mathcal{L}_r$ increased with the increasing $\lambda_i$ while $\lambda_r$ was fixed to one. It was considered that higher weight parameters $\lambda_i$ for the initial and boundary conditions minimized the emphasis on the residual loss term $\mathcal{L}_r$ (see Fig. 7 (**k**) and (**j**)). The increased $\mathcal{L}_r$ lead to less accurate solutions or higher $\epsilon^\theta$, which is evident in Fig. 7 (**m**) and (**j**). These complicated trends make the PINN approach less consistent than traditional numerical methods. Note that automatic but empirical tuning of $\lambda_i$ proposed by Wang et al. (2021) did not improve the results in our case, particularly with the L-LAAF, which is shown in Fig. S2. Because

finding the optimal values for $\lambda_i$ is not our primary purpose, we use the value of ten for all the three weight parameters $\lambda_i$ for the following analysis.

### 3.1.6   Effects of number of residual and boundary data points

The effects of the number of residual points $N_r$, where the residual of the RRE is evaluated, were investigated by varying the number $N_r \in \{1000, 3000, 10000, 30000, 100000\}$, which resulted in 50 runs in total. NN architecture was fixed to 5 layers

with 50 units with the L-LAAF. As expected, larger error was observed when smaller number of residual points was used (see Fig. 8 (**a**)). However, even if we increased the number to 30000 and 100000, the performance of PINNs did not improve. We concluded that this was due to the simultaneous but opposite effect of the number $N_r$ on the loss term $\mathcal{L}_r$, as shown in Fig. 8 (**b**). This was because increasing $N_r$ is equivalent to minimizing the importance of each residual point randomly selected in the spatial and temporal domain. Note that we tested the residual-based adaptive refinement algorithm proposed by Lu et al.

(2021b), where additional residual points are added while training NNs based on the distribution of the residual values. As shown in Fig. S1 (**a**), the algorithm seemed to improve the performance of PINNs, but the effectiveness was minor. These findings demonstrate the difficulty in finding an optimal strategy to distribute residual points for PINNs to learn solutions to PDEs with high accuracy.

Also, the effects of the number of upper boundary data points $N_{ub}$ were studied, where $N_{ub} = \{100, 300, 1000, 3000, 10000\}$.

NN architecture and training algorithms were set to the same as Sect. 3.1.2. Figure 9 (**a**) showed that more than 300 upper boundary data points $N_{ub}$ are necessary for PINNs to learn the solution well. This is because PINNs required enough upper boundary data points to capture the surface flux in particular near the initial condition. However, increasing $N_{ub}$ from 300 did not improve the performance of PINNs. At the same time, we observed the increase in the loss term for the upper boundary condition $\mathcal{L}_{ub}$, as shown in 9 (**b**). This observation is similar to the case for the residual points and makes it difficult to

determine optimal $N_{ub}$. The difficulty in tuning parameters for PINNs is further explained in the next section.

### 3.1.7   Toward more consistent performance of PINNs

We demonstrated that PINNs can approximate the solution to the RRE with accuracy comparable to the FDM. However, a significant amount of effort was needed to tune the NN architecture and parameters, and those optimal settings depend on problems of interest, which makes it very challenging for PINNs to be consistent numerical solvers of PDEs. To understand

why the performance of PINNs is not consistent, we investigated the relationships between $\mathcal{L}_i$ and $\epsilon^\theta$ by compiling the results





**Figure 7.** Homogeneous soil. The effects of the weight parameters in the loss function (Eq. 17) for the initial condition $\lambda_{ic}$, the lower boundary condition $\lambda_{lb}$, and the upper boundary condition $\lambda_{ub}$ on the loss terms $\mathcal{L}_{ic}$, $\mathcal{L}_{lb}$, $\mathcal{L}_{ub}$, $\mathcal{L}_r$ for the initial and boundary conditions and the residual of the PDE, respectively, and the relative squared error in terms of volumetric water content $\epsilon^\theta$. $\lambda_r$ was fixed to one.



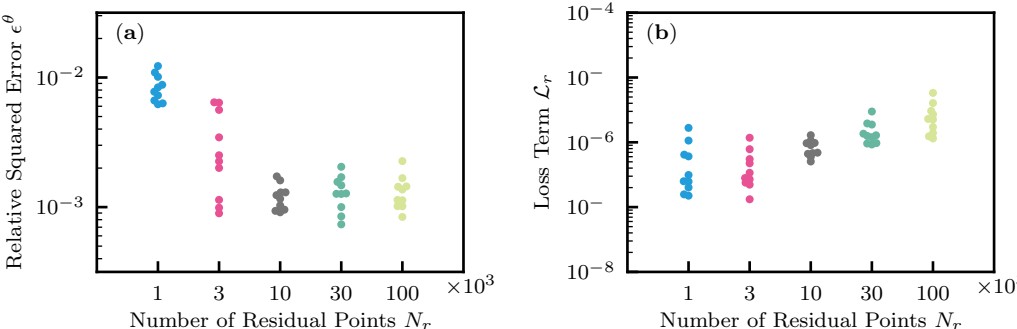

**Figure 8.** Homogeneous soil. (**a**): The effect of the number of residual points $N_r$ on the relative squared error $\epsilon^\theta$. (**b**): The effect on the loss term for the residual $\mathcal{L}_r$.

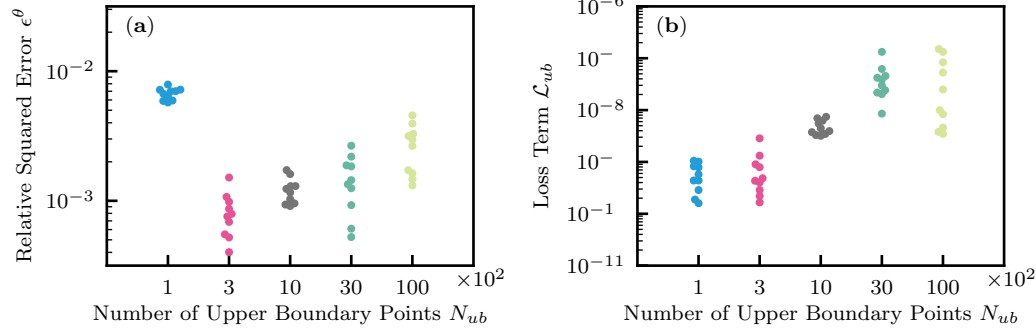

**Figure 9.** Homogeneous soil. (**a**): The effect of the number of upper boundary data $N_{ub}$ on the relative squared error $\epsilon^\theta$. (**b**): The effect on the loss term for the upper boundary condition $\mathcal{L}_{ub}$.

from the previous sections. The left column of Fig. 10 corresponds to a fixed NN (5 hidden layers with 50 units with the L-LAAF), and the center column is for NNs with varying architecture (the number of hidden layers and units) with the L-LAAF (from Sect. 3.1.4), while the right column is for a fixed NN with varying weight parameters $\lambda_i$ in the loss function (from Sect. 3.1.5). Note that the dimension of the loss function (i.e., the number of adjustable parameters) is the same for the first and third

columns because the NN architecture is the same, while the shape or landscape of the loss function is different because of the varying weight parameters $\lambda_i$.

    As for the first column, the PINN solution was consistent regardless of the random seeds used for the NN initialization. Although there were some differences in the accuracy, a detailed examination of the PINN solutions revealed that the errors mainly came from near the upper boundary. Thus, we concluded that we obtained consistent PINN solutions for this problem

once we determined the NN architecture. However, determining NN architecture is still empirical and depends on problems. Based on other studies and our experiences, NNs consisting of less than ten hidden layers appear to be enough to approximate





the solution to PDEs. However, the application of PINNs to PDEs with large spatial and temporal domains requires more investigation.

Figure 10 (**k**) and (**l**) demonstrates that the residual loss $\mathcal{L}_r$ is well correlated with $\epsilon^\theta$, which coincides with the theoretical study by Mishra and Molinaro (2022). This observation might indicate that smaller residual loss $\mathcal{L}_r$ is more important than other loss terms. If this speculation is true, this implies the possibility of transfer learning, where NNs are pre-trained with only the residual of PDEs without initial and boundary conditions and later fine-tuned by them, which could drastically reduce the computational work and needs more investigation.

## 3.2 Heterogeneous Soil

In this section, we simulated a one-dimensional infiltration into a two-layered soil with a length of 20 cm. Because each layer has a distinct saturated hydraulic conductivity $K_s$, the solution to the RRE is not differentiable at the boundary of the layers. Thus, we implemented the domain decomposition method (see Sect. 2.4.4) by dividing the spatial domain into the upper domain $\Omega_U$ ($-10 \leq z \leq 0$ cm) and the lower domain $\Omega_L$ ($-20 \leq z \leq 10$ cm). NNs $\mathcal{N}_U(z,t;\Theta_U)$ and $\mathcal{N}_L(z,t;\Theta_L)$ were assigned to $\Omega_U$ and $\Omega_L$, respectively. The two NNs interact with each other through the interface conditions described in Sect. 2.4.4 and were trained simultaneously, although separate training is also possible, as in Jagtap and Karniadakis (2020).

We compared the PINN solution with an FEM solution obtained by HYDRUS-1D (Šimůnek et al., 2013). FEMs are similar to PINNs in that both methods use some basis functions to approximate the solution of PDEs. While PINNs use NNs as the basis function, the FEM implemented in HYDRUS-1D uses a linear finite element as the basis function. Although HYDRUS-1D implements a variable time step, we used a constant time for the comparison. Because WRCs and HCFs defined by Eq. 5 and 6 are not implemented in HYDRUS-1D, we used a lookup table feature to provide HUDRUS-1D with the WRC and HCF manually.

### 3.2.1 Problem Setup

WRCs and HCFs relationships for the soils are the same for the homogeneous case. The saturated conductivity $K_s$ is 10.0 and 1.0 cm s$^{-1}$ for the upper layer (from $z = -10$ cm to $z = 0$ cm) and the lower layer (from $z = -20$ cm to $z = -10$ cm), respectively. Other parameters $\theta_s$, $\theta_r$, and $\alpha$ for the two layers as well as the initial and boundary conditions are the same as the homogeneous case.

### 3.2.2 Characteristics of PINN solution

Figure 11 (**a**) shows the PINN solution with the analytical solution introduced in Sect. 2.2.2. Both NNs $\mathcal{N}_U$ and $\mathcal{N}_L$ consisted of 5 hidden layers with 50 units with the L-LAAF, and $\beta$ was set to one. Randomly sampled 10000 residual points and equally spaced 101 initial data points were used for both NNs. The upper and lower boundary conditions were given as the case for the homogeneous soil. To connect the two NNs, randomly sampled 1000 points were used for the three interface continuity conditions: the water flux $\mathcal{L}_{I_q}$ (Eq. 27); the residual $\mathcal{L}_{I_r}$ (Eq. 28); the NN output $\mathcal{L}_{I_\mathcal{N}}$ (Eq. 29). All the weight parameters



**Figure 10.** Homogeneous soil. The relationships between loss terms $\mathcal{L}_i$ and the relative squared error in terms of volumetric water content $\epsilon^\theta$ for a fixed neural network (NN) with different NN initializations (left column), NNs with varying architecture (center column), and a fixed NN with varying weight parameters $\lambda_i$ in the loss function (right column). $\mathcal{L}_{ic}$, $\mathcal{L}_{lb}$, $\mathcal{L}_{ub}$, and $\mathcal{L}_r$ are the loss terms for the initial condition, lower and upper boundary conditions, and the residual of the PDE, respectively.





in the loss function $\lambda_i$ were set to ten while $\lambda_r$ for the lower layer was set to one. Figure 11 (**b**) showed the FEM solution obtained using HYDRUS-1D with $dz = 0.1$ cm and $dt = 0.01$ h, which is comparable to the temporal resolution of the upper

boundary data points given to the PINNs. The PINN solution was superior to the HYDRUS-1D solution ($\epsilon^\theta = 3.99 \times 10^{-3}$ for the PINN and $\epsilon^\theta = 1.67 \times 10^{-2}$ for the FEM solution, respectively), while the HYDRUS-1D solution underestimated the volumetric water content in the upper layer near the boundary, which coincides with Brunone et al. (2003). This is because only the matric potential continuity is guaranteed in HYDRUS-1D, while both matric potential and water flux continuity conditions were imposed for the PINNs. Also, HYDRUS-1D consistently overestimated the volumetric water content at the wetting front

in the lower layer, while consistent errors were not observed for the PINN solution. From this comparison, we concluded that PINNs with domain decomposition can approximate the solution of the RRE for a two-layered soil with discontinuous hydraulic conductivity.

### 3.2.3 Training PINNs

The left column of Fig. 12 shows the evolution of the loss terms. All the loss terms for both layers remained higher than those

for the homogeneous case (see Fig. 4), which demonstrated the difficulty in training PINNs for the layered soil case. While the Adam algorithm resulted in a well-approximated solution for the homogeneous case, the L-BFGS-B algorithm played an important role for the heterogeneous case, particularly in reducing the loss term for the upper boundary condition $\mathcal{L}_{ub}$ and the residual $\mathcal{L}_r$ for the upper layer. Figure 12 (**c**) illustrated that the three interface conditions were satisfied well.

The right column of Fig. 12 shows how the adaptive parameters $sa$ for the L-LAAF changed during the training. We

expected $sa$ for the lower layer to be similar to the homogeneous case because the solution for the lower layer is similar to the homogeneous case. However, Fig. 12 (**b**) showed that $sa$ for the lower layer was much smaller than that for the homogeneous case while similar trends of $sa$ for different hidden layers were observed (i.e., layer 2 was the highest). $sa$ for all the layers of both layers reached their limiting values after approximately 20000 to 30000 iterations of Adam algorithm, which coincided with the homogeneous case and Jagtap et al. (2020). This indicated that if we can find a better initial guess of $sa$ for each layer,

we may be able to speed up the training of PINNs, which requires further research.

Figure 13 demonstrated how PINNs learned the solution. At the initialization (Fig. 13 (**a**)), there is discontinuities at the boundary, which is evident for $t = 10$ h. The continuity conditions and the lower boundary condition were quickly met. The PINNs started to capture the flow of soil moisture at the 20000 iterations of the Adam algorithm (Fig. 13 (**e**)), which coincided with when the adaptive parameters $sa$ for the L-LAAF reached their limiting values (see the right column of Fig. 12). Even

at the end of the Adam algorithm, there are large errors in the PINN solution near the surface and wetting fronts in the lower layer (Fig. 13 (**g**)). Those errors were further minimized by the L-BFGS-B algorithm (Fig. 13 (**h**)). This demonstrated that a second-order method such as the L-BFGS-B algorithm is necessary to train PINNs when the solution to PDEs is complicated.

### 3.2.4 Effects of number of interface points and weight parameters in loss function

We investigated the effects of the number of interface points $N_I$ on PINN performance. The number varied from 100, 300,

1000, 3000, and 10000, and ten different random seeds were used for each setting. Figure 14 (**a**), (**b**), and (**c**) showed that the



**(a)** Physics-informed neural networks

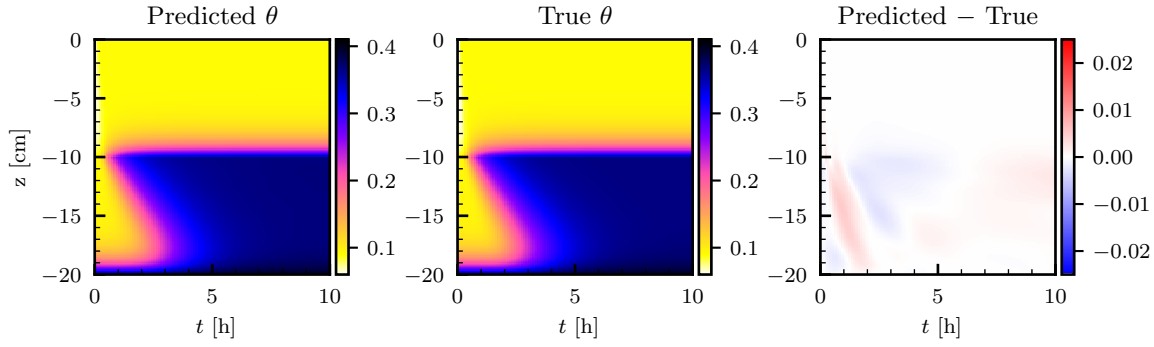

**(b)** Fintie element method (HYDRUS-1D)

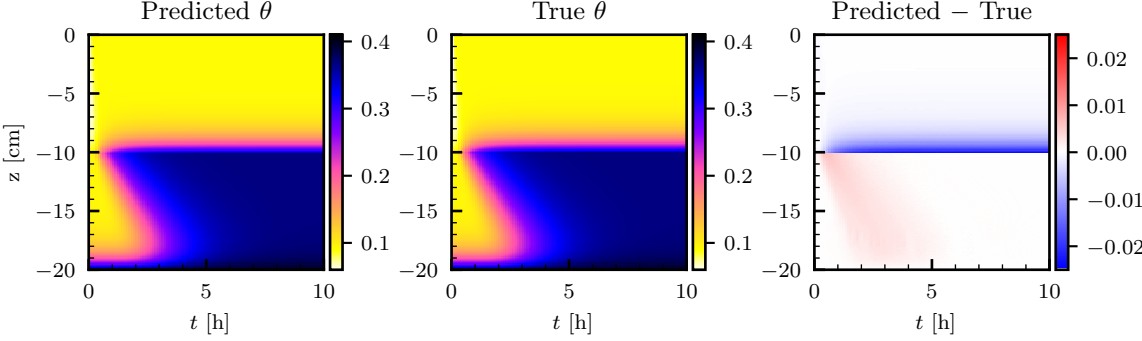

**Figure 11.** Heterogeneous soil. The saturated conductivity $K_s$ is 10.0 and 1.0 cm s$^{-1}$ for the upper and lower layer, respectively. (**a**): Physics informed neural network (PINN) solution in terms of volumetric water content $\theta$ [-] obtained by two neural networks of 5 hidden layers with 50 units with the layer-wise adaptive activation function (left column). True analytical solution (center column) is given by Srivastava and Yeh (1991) (see Sect. 2.2.2), and the difference between the PINN and true solutions are shown in the right column. (**b**): Numerical solution by a finite element method was obtained with a spatial mesh of $dz = 0.1$ cm and a time step $dt = 0.01$ h using HYDRUS-1D (Šimůnek et al., 2013).

**(a)** Upper layer

**(b)** Lower layer

**(c)** Interface

**Figure 12.** Heterogeneous soil. **(a)**: The evolution of the loss terms in the loss function (left column) and adaptive parameters $sa$ for the layer-wise locally adaptive activation function (Eq. 9) for each hidden layer (right column) during the Adam (100000 iterations) and the following L-BFGS-B training for the neural network for the upper layer. Here, Layer 1 is next to the input layer. **(b)** Those for the lower layer. **(c)**: The evolution of the loss terms for the interface conditions.



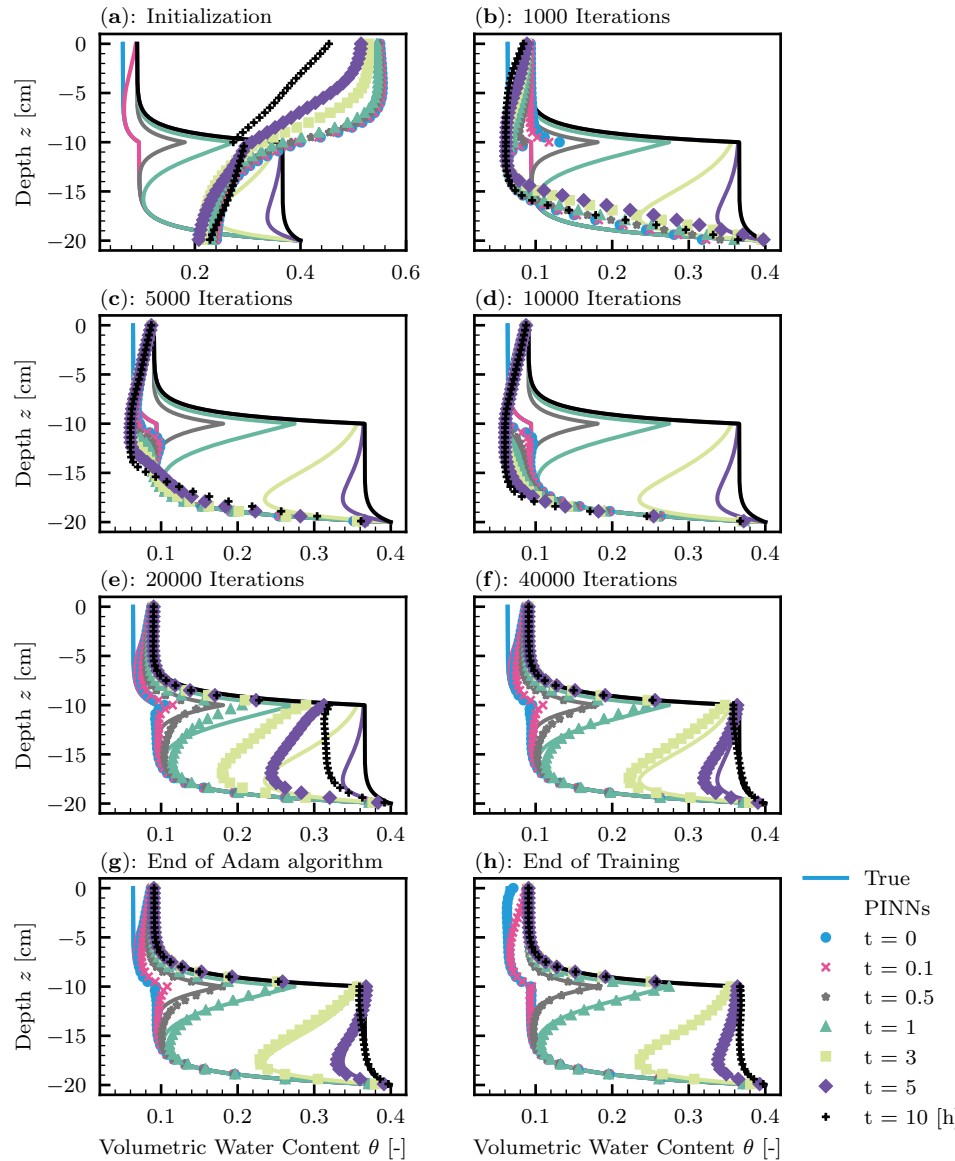

**Figure 13.** Heterogeneous soil. The evolution of the PINN solution during the training. (**a**): Initialization of the PINNs. (**b**) to (**f**): 1000, 5000, 10000, 20000, 40000 iterations of the Adam algorithm. (**g**): The end of 100000 iterations of the Adam algorithm. (**h**): The end of the L-BFGS-B algorithm.



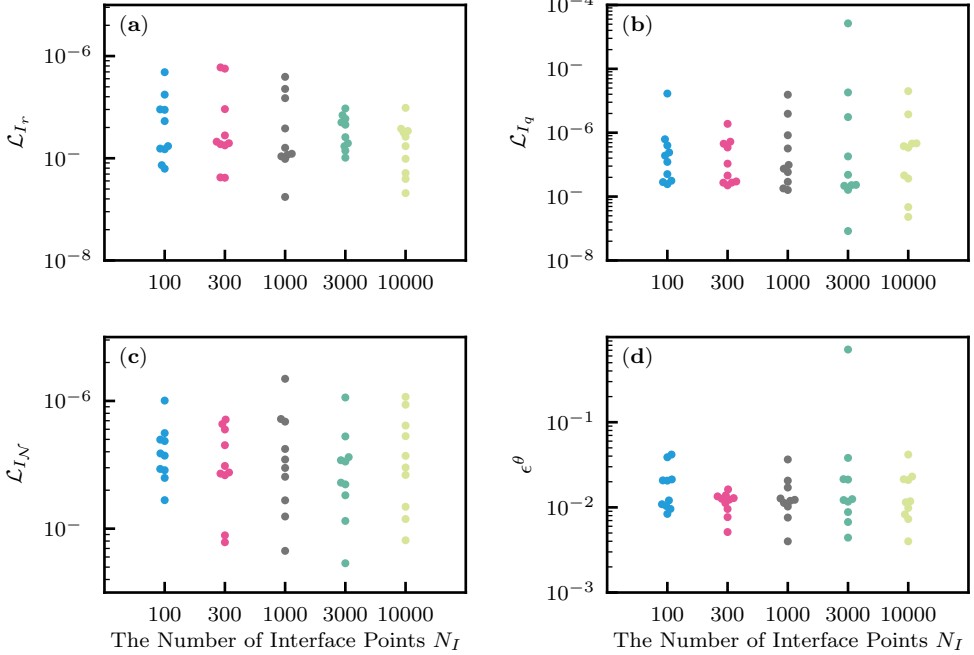

**Figure 14.** Heterogeneous soil. The effects of the number of interface data points $N_I$. (**a**): The loss term for the continuity of the residual $\mathcal{L}_{I_r}$. (**b**): The loss term for the continuity of the water flux $\mathcal{L}_{I_q}$. (**c**): The loss term for the continuity of the the neural network output $\mathcal{L}_{I_\mathcal{N}}$. (**d**): The relative squared error with respect to the volumetric water content $\epsilon^\theta$.

effects on the loss terms for the interface conditions were negligible. The relative squared error $\epsilon^\theta$ decreased with the number $N_I$ from 100 to 300, but the effect was negligible for larger $N_I$ (see Fig. 14 (**d**)). Thus, we concluded that the effects $N_I$ were minor and used 1000 interface points in the following analysis.

We also investigated the effects of the weight parameters $\lambda_i$ in the loss function. We fixed $\lambda_r$ for the lower layer to be one and $\lambda_i$ for the initial condition and the lower boundary condition for the lower layer to be ten while $\lambda_i$ for the upper layer and the interface conditions were varied from 1, 10, and 100 (i.e., $\lambda_i$ for the different loss terms in the upper layer are the same). Ten different initializations were conducted. Figure 15 (**a**) shows PINNs were more likely trapped by a local minimum of the loss function when $\lambda_i$ for the upper layer was smaller, indicated by the cloud of the data points. However, the best PINN solution appeared not to be affected by $\lambda_i$. Similar observations were made for the effects on the loss terms corresponding to the upper layer (see Fig. S4). Figure 15 (**b**) illustrated the effects of $\lambda_i$ for the interface conditions. When $\lambda_i$ were larger, the PINNs produced worse solutions, and it was evident that PINNs suffered from a local minimum of the loss function. Similar conclusions were made from the effects on the loss terms for the upper layer (see Fig. S4). These observations led us to conclude that choosing the right weight parameters $\lambda_i$ in the loss function is very important and challenging for the heterogeneous case to achieve accurate and consistent solutions to the RRE.





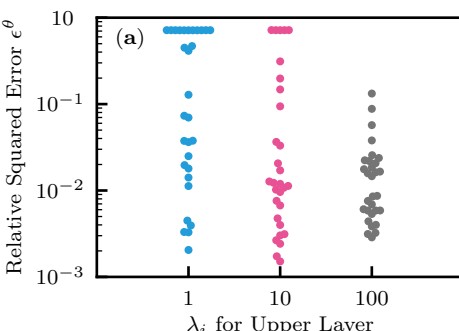
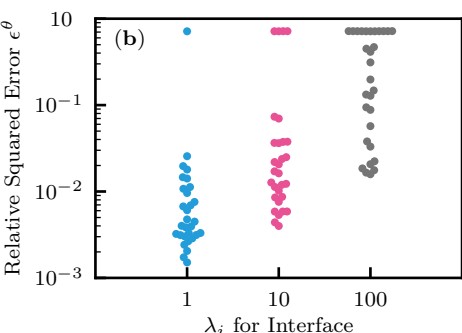

**Figure 15.** Heterogeneous soil. The effects of weight parameters $\lambda_i$ in the loss function on the performance of PINNs with respect to the relative squared error of the volumetric water content $\epsilon^\theta$. (**a**): Upper layer. (**b**): Interface conditions.

## 4 Inverse modeling

In this section, we demonstrate that inverse modeling can be easily implemented using PINNs. Here, we aim to estimate a surface water flux (i.e., upper boundary condition) from near-surface moisture measurements in a layered soil. Surface water flux is the result of precipitation, evaporation, and surface runoff and thus essential information for land surface modeling and groundwater management. Although rainfall measurements can be used as surface water flux, the measurements are generally spatially scarce and noisy. Therefore, it is important to estimate surface water flux from near-surface soil moisture data. In this line of research, Sadeghi et al. (2019) employed the analytical solution of the linearized RRE, and Li et al. (2021a) proposed a deterministic inverse algorithm to estimate surface water flux given past surface water flux. Brocca et al. (2013) used a simple soil water balance equation to estimate rainfall. These studies assumed soil hydraulic properties are homogeneous (Labolle and Clausnitzer, 1999). Here, we present an inverse framework based on PINNs to estimate surface water flux from near-surface soil moisture measurements in a two-layered soil, as an extension of our previous work (Bandai and Ghezzehei, 2021).

### 4.1 Problem setup

We consider a one-dimensional soil moisture dynamics in a layered soil. The upper layer is a 10 cm depth of loam soil (0 to -10 cm), and the lower layer is a 10 cm depth of sandy loam (-10 to -20 cm). The WRCs and HCFs of the soils for $\psi < 0$ are represented by the van-Genuchten Mualem model (Mualem, 1976; van Genuchten, 1980):

$$\theta = \theta_r + \frac{\theta_s - \theta_r}{(1 + (-\alpha_{VG}\psi)^{n_{VG}})^m}, \tag{31}$$

$$K = K_s S_e^l (1 - (1 - S_e^{1/m})^m)^2, \tag{32}$$

where $\theta_r$ is the residual water content [L$^3$ L$^{-3}$]; $\theta_s$ is the saturated water content [L$^3$ L$^{-3}$]; $\alpha_{VG}$ [L$^{-1}$] and $n_{VG}$ [-] determine the shape of the WRC; $K_s$ is the saturated hydraulic conductivity [L T$^{-1}$]; $l$ is the tortuosity parameter; $m = 1 - 1/n_{VG}$; $S_e$


is the effective saturation defined as

$$S_e := \frac{\theta - \theta_r}{\theta_s - \theta_r}. \tag{33}$$

The parameters for the two soils were set in the following way: $\theta_r = 0.078$, $\theta_s = 0.43$, $n_{VG} = 1.56$, $K_s = 1.04$, and $l = 0.5$ for the loam soil (upper layer); $\theta_r = 0.065$, $\theta_s = 0.41$, $n_{VG} = 1.89$, $K_s = 4.42$, and $l = 0.5$ for the sandy loam soil (lower layer). The lower boundary is a constant pressure head set to $\psi = -1000$ cm, and the upper boundary condition is a variable surface water fluxes as follows: $-0.3$ cm h$^{-1}$ from $t = 0$ to $t = 8$ h; $0.02$ cm h$^{-1}$ from $t = 8$ to $t = 12$ h; $-0.2$ cm h$^{-1}$ from $t = 12$

to $t = 20$ h. The positive and negative values represent evaporation and infiltration, respectively. The initial condition was set to $\psi = -1000$ cm for all the depths. To generate synthetic data for the abovementioned scenario, we employed HYDRUS-1D (Šimůnek et al., 2013) to compute the numerical solution of the RRE with $dt = 0.0001$ h and $dz = 0.02$ cm. The numerical solution by HYDRUS-1D is not necessarily accurate because it may contain undesirable numerical errors near the interface, as observed in Section 3.2 although we confirmed the global mass balance of the solution. We further added a Gaussian noise

with a mean of 0 and a standard deviation of 0.005 to the numerical solution. We sampled the simulated noisy synthetic data at predetemined locations to mimic soil moisture measurements by in-situ sensors. We tested three patterns of the measurement locations $z_m$ [cm]: $z_m \in \{-5, -15\}$; $z_m \in \{-3, -7, -13, -17\}$; $z_m \in \{-1, -5, -9, -13, -17\}$. The temporal resolution of the measurements was 0.1 h.

### 4.2 PINNs inverse solution

To infer the surface water flux upper boundary condition, we constructed PINNs with domain decomposition. The two NNs consisted of 5 hidden layers with 50 units, as in Sect. 3.2, and $\beta = 0$ was used for the output of both NNs (Eq. 15). Unlike the forward modeling, the initial and boundary data points were not used, and the sampled synthetic data points and randomly sampled collocations points were only used to train the NNs. Therefore, the loss function is the sum of the loss term for the measurement data $\mathcal{L}_m$, the residual of the RRE $\mathcal{L}_r$, and the three interface conditions ($\mathcal{L}_{I_r}$, $\mathcal{L}_{I_q}$, and $\mathcal{L}_{I_N}$). As for the weight

parameters $\lambda_i$ for each loss term, $\lambda_i = 10$ for the measurement data for both layers and the residual loss for the upper layer and $\lambda_i = 1$ for the interface conditions, while $\lambda_i = 1$ for the residual loss for the lower layer as a reference. We tested ten different NN initializations for each measurement scheme, and thus a total of 30 simulations were conducted. Note that the surface water flux $i(t)$ was estimated by evaluating Eq. 14 with the solution of the RRE by PINNs.

Figure 16 showed the best-recovered solution and estimated surface upper boundary condition for the three measurement

schemes. As expected, more accurate recovered solutions were obtained from dense soil moisture measurements (see also Fig. S7). However, interestingly, NNs more likely were trapped by bad solutions when more measurement data were given (see Fig. S7). This means a large amount of data does not necessarily lead to a good performance of PINNs because large data make the training of PINNs more difficult. This is a practically important point and requires further investigation. As for the two measurement location scheme (Fig. 16 (**a**)), the wetting front reached the top measurement point ($z_m = -5$ cm) at

approximately $t = 3$ h, which coincided with when the estimated surface water flux $i$ was reasonable. After that time, both the recovered solution and estimated surface water flux were quite reasonable. Similar trends were observed for the other





two measurement schemes (Fig. 16 (**b**) and (**c**)). This suggested that we need soil moisture measurement data closer to the surface ($z = 0$ cm) to capture the wetting front and the infiltration rate. Figure S8 in the supplementary material showed the evolution of the loss terms $\mathcal{L}$ corresponding to the measurement data, residual, and the interface conditions. Although the direct

comparison between the forward and inverse modeling is not possible because the problem settings are different, we observed smaller residual loss terms $\mathcal{L}_r$ for the inverse modeling. This observation and our experiences indicate that PINNs are more effective for the inverse problem than the forward modeling because data points inside the spatial and temporal domain are more informative than initial and boundary conditions for NNs to find the solution to PDEs.

We note that soil hydraulic parameters are known for both layers in this test, which is not the case for field applications.

Depina et al. (2021) implemented PINNs with a global optimization algorithm to estimate the van-Genuchten parameters of a homogeneous soil ($K_s, \alpha_{VG}$, and $n_{VG}$) from soil moisture measurements, and the framework was tested against for both synthetic and laboratory infiltration experiment data. They demonstrated that PINNs with a global optimization algorithm could determine the van-Genuchten parameters for a homogeneous soil. In fact, the current study was motivated by the need to verify a PINN approach to estimate such soil hydraulic parameters of a layered soil for field applications. Our next research objective

is to implement PINNs that can estimate both the upper surface boundary condition and soil hydraulic parameters (e.g., van-Genuchten parameters) for layered soils and test them with soil moisture and surface water flux data measured in a lysimeter. Note that the estimation of surface water flux was reasonable when the value is positive (i.e., evaporation) in the example, but it would probably not be the case for field applications because evaporation requires coupled heat and water transport models. Applying PINNs to multi-physics in unsaturated hydrology is also our next research step.

## 5 Advantages and disadvantages of PINNs

Regardless of the potential of PINNs to solve PDEs, several studies reported their failures and limitations (Fuks and Tchelepi, 2020; Sun et al., 2020; Wang et al., 2021). Although there are some theoretical studies on the convergence and error analysis on PINNs (e.g., Mishra and Molinaro, 2022; Shin et al., 2020), theoretical understanding of PINNs is still in its infancy (Karniadakis et al., 2021). We summarize the advantages and disadvantages of PINNs compared to traditional numerical

methods (e.g., finite difference, finite element, and finite volume methods) to potentially use the method to solve essential questions in hydrology, including large-scale forward and inverse modeling.

One main drawback of PINNs for forward modeling is their computational time. For our case studies, it took approximately 30 and 90 min for the homogeneous and heterogeneous forward modeling using a desktop computer with GPU (NVIDIA GeForce RTX 2060), while it took less than 1 min for HYDRUS-1D to solve the heterogeneous problem. PINNs might be

more competitive for large-scale hydrology problems, which needs further investigation.

For forward modeling, PINNs can treat initial and boundary conditions as data points. This feature is advantageous over traditional numerical methods because the accurate initial and boundary conditions are virtually impossible to obtain in practical conditions. On the other hand, traditional methods can take into account the uncertainties of the initial and boundary conditions through the Bayesian approach while it requires solving the forward problem many times. Although uncertainty





**(a)** Two measurement locations

**(b)** Four measurement locations

**(c)** Five measurement locations

**Figure 16.** Inverse modeling to estimate surface water flux from soil moisture measurements in a layered soil (upper layer: loam soil; lower layer: sandy loam soil). True solution generated by HYDRUS-1D and the recovered solution by PINNs (left column) and the true and estimated upper surface water flux boundary condition (right column) for different measurement locations $z_m$ [cm]. **(a)**: $z_m \in \{-5, -15\}$. **(b)**: $z_m \in \{-3, -7, -13, -17\}$. **(c)**: $z_m \in \{-1, -5, -9, -13, -17\}$.





quantification through PINNs is open and challenging questions (Psaros et al., 2022), the capability of PINNs to deal with noisy and incomplete initial and boundary conditions is noteworthy.

    Traditional numerical methods require mesh generation, which can be tedious when the spatial domain is complicated. On the other hand, PINNs can be easily modified to accommodate such complicated geometries (Raissi et al., 2020). However, it is challenging to correctly impose boundary conditions on PINNs while they are imposed softly in the loss function in this

study. Regarding hydrology application, system-dependent boundary conditions such as ponding and evaporation conditions are challenging to implement because PINNs solve PDEs in spatial and temporal domains simultaneously, rather than sequentially, as in traditional time-stepping methods. This difficulty may be a technical issue, but the loss function would be more complicated with such system-dependent boundary conditions, and thus training NNs would be more difficult.

    One of the main challenges of PINNs is training PINNs for large-scale modeling. In particular, a long-term simulation such

as wetting and drying cycles requires PINNs to approximate very complicated functions. We may sequentially train PINNs in time but lose the ability of PINNs to solve PDEs simultaneously in space and time, and numerical and optimization errors accumulate with time stepping.

    The application of PINNs to multi-scale and multi-physics problems is currently challenging, although there are some pioneering studies in hydrology (He et al., 2020). It is known that the solution of PINNs to a multi-scale problem is not

always accurate, even for simple problems, because NNs tend to learn "easy" or low-frequency parts of the solution (Wang et al., 2022). Although the study was only concerned with water flow in unsaturated soils, near-surface soil moisture dynamics is essentially coupled heat and water transport. Therefore, further research is needed for the application of PINNs to multi-physics simulations in unsaturated soils.

    One advantage of PINNs specific to the RRE is that there is no need for temporal discretization, which results in mass

balance issue (Celia et al., 1990). Also, PINNs solutions are differentiable and thus can be used to derive water flux easily without post-processing, as in Scudeler et al. (2016). Furthermore, PINNs can store the solutions of PDEs efficiently with a smaller number of degrees of freedom, particularly for high dimensions (Karniadakis et al., 2021). Those merits can make PINNs a good candidate for a numerical solver of large-scale modeling based on the RRE. Nevertheless, these advantages depend on how accurately PINNs satisfy the RRE.

In terms of inverse modeling, PINNs have some interesting features. First, PINNs do not have to solve the forward modeling to solve the inverse problem. On the other hand, standard inverse methods require solving the forward modeling many times to adjust parameters of interest. This feature makes the computation of PINNs efficient. However, as shown in the study, PINNs do not precisely impose PDEs constraints as traditional methods, where the forward modeling is actually solved. Further research is needed to minimize the residual loss term so that known physics is precisely imposed. Second, when estimating boundary

conditions as in the study or initial condition from data, PINNs do not require the discretization of those target functions. In traditional methods, it is common to represent the target functions as a linear combination of some basis functions (e.g., finite elements) and estimate the coefficients of the basis functions. In PINN framework, such discretization is not necessary, and those target function values can be evaluated directly from NNs. Overall, although PINNs have interesting and attractive





characteristics, fully utilizing the potential requires further research. In the next section, future perspectives of PINNs are

mentioned.

## 6    Conclusions and future perspectives

We presented a numerical method based on neural networks (NNs), called physics-informed neural networks (PINNs), to solve

the Richardson-Richards equation (RRE) to simulate water flow in unsaturated homogeneous and heterogeneous soils. We

tested recently proposed PINN algorithms on our problems and found that the layer-wise locally adaptive activation function

(L-LAAF) developed by Jagtap et al. (2020) was effective. The L-LAAF changes the slope and the linear regime of the

activation functions in NNs and helps PINNs approximate the solution of the RRE well. First, we tested the PINN approach

for the homogeneous soil case. By comparing the PINN solution to the analytical solution by Srivastava and Yeh (1991) and

the numerical solution by a finite difference method, we demonstrated that "well-trained" PINNs can be competitive in terms

of accuracy and memory efficiency. However, training PINNs requires significant efforts to tune various parameters of NNs,

including NN architecture and weight parameters in the loss function. We systematically investigated the effects of those

parameters on the performance of PINNs and demonstrated that those interrelated effects make PINN approach less consistent.

Although some automatic but empirical algorithms to tune those parameters improved the performance of PINNs to some

extent, it was difficult for PINNs to consistently obtain solutions to the PDE with high accuracy, and the results were strongly

dependent on the initialization of NNs. Our empirical but comprehensive observations provide some suggestions on the choice

of the parameters, but we do not think they can be applied to various cases, and thus further studies are necessary.

We tested PINN approach for a layered soil, where hydraulic conductivity is discontinuous across the layer boundary. The

analytical solution by Srivastava and Yeh (1991) was used to verify the PINN solution. We demonstrated that PINNs with

domain decomposition proposed by Jagtap and Karniadakis (2020) successfully approximated the solution of the RRE for a

two-layered soil. The comparison with a finite element method using popular software, HYDRUS-1D (Šimůnek et al., 2013),

was made. The PINN solution was superior to HYDRUS-1D for the problem because the interface conditions on the layer

boundary were well imposed for the PINN approach but not for HYDRUS-1D. Nevertheless, the study demonstrated that

obtaining PINN solutions for the problem with consistent accuracy was challenging because of the difficulty in choosing the

right weight parameters in the loss function, which determines the relative importance of physical constraints for the problem

(e.g., initial and boundary conditions).

We further applied the PINNs with domain decomposition to the inverse modeling to estimate a water flux upper boundary

condition from noisy sparse soil moisture measurements. The inverse modeling was easily formulated by the PINN approach,

and the effects of the measurement schemes were studied. The upper boundary condition was reasonably inverted from the

noisy data, in particular when measurement data near the soil surface were available. However, our results demonstrated that

a large amount of data do not necessarily lead to a good performance of PINNs because training PINNs is more difficult with

more data. Further research is needed to make PINNs learn from a larger amount of data and simultaneously determine both

soil hydraulic properties and surface water flux for layered soils.



PINNs have the potential to solve issues traditional numerical methods cannot solve by leveraging the capability of NNs to approximate complex functions efficiently. However, the mathematical complexities of the forward and inverse problems are lumped into a complicated non-linear, non-convex minimization problem. Whether PINNs can perform well depends

on whether we can solve the resulting minimization problem well. Another difficulty comes from the fact that PINNs have an unusual regularization term in the loss function as the form of the residual of PDEs. This term is very different from standard regularization terms such as L1 and L2 regularizations because it contains the derivative of the output of NNs with respect to their input. The mathematical and exploratory investigation of the minimization problem and the regularization term is necessary for further improvements of PINNs. The investigation may include a vast amount of literature on NNs and

PDE constrained optimization. There are many methods and findings that have not been well tested against PINNs, including transfer learning, second-order optimization methods, and the correspondence with adjoint-state methods. We will investigate those areas to improve the understanding of PINNs and use PINNs for large-scale modeling in hydrology.

Aside from PINNs, the latest research trends have been directed toward learning the "operator" of PDEs rather than their solutions given initial and boundary conditions, as in the study. This new research field has been led by two main groups

(Lu et al., 2021a; Li et al., 2021b), and they aim to develop operator learning methods applicable to general PDEs, of course including the ones in hydrology. Do we just wait until they develop general PDE simulators or provide a unique perspective in soil physics and hydrology? We need to consider how we contribute to the rapid progress of the fields as domain scientists (Nearing et al., 2021).

*Code and data availability.* All the Python codes and data to reproduce the results are available on Bandai and Ghezzehei (2022)

**Appendix A: List of abbreviations**

FDMs: Finite Difference Method

FEMs: Finite Element Methods

GPUs: Graphics Processing Units

HCF: Hydraulic Conductivity Function

L-LAAF: Layer-wise Locally Adaptive Activation Function

ML: Machine Learning

NNs: Neural Networks

PDE: Partial Differential Equation

PINNs: Physics-Informed Neural Networks

RRE: Richardson-Richards Equation

WRC: Water Retention Curve



### Appendix B: List of notations

**superscript**

$\hat{\cdot}$ : prediction except for used for $\hat{\Theta}$

$\cdot^*$: dimensionless for the analytical solutions

**subscript**

$\cdot_D$: Dirichelt boundary condition

$\cdot_F$: water flux boundary condition

$\cdot_I$: interface

$\cdot_{I_\mathcal{N}}$: interface condition regarding the continuity of neural network output

$\cdot_{I_q}$: interface condition regarding the continuity of water flux

$\cdot_{I_r}$: interface condition regarding the continuity of the residual of the RRE

$\cdot_{I_\psi}$: interface condition regarding the continuity of water potential

$\cdot_{ic}$: initial condition

$\cdot_{lb}$: lower boundary condition

$\cdot_L$: lower layer

$\cdot_m$: measurement data

$\cdot_r$: residual

$\cdot_{ub}$: upper boundary condition

$\cdot_U$: upper layer

**alphabet**

**a**: the collection of trainable parameters for adaptive activation functions for a neural network

$a^{[k]}$: trainable parameter for the element-wise non-linear activation function

**b**: the collection of bias vectors for a neural network

$\mathbf{b}^{[k]}$: bias vector for the $k$th hidden layer

$dt$: time step for finite difference and finite element solutions [T]

$dz$: spatial mesh for finite difference and finite element solutions [L]

$g(z)$: initial condition

$h(z,t)$: Dirichlet boundary condition

$H$: total water head [L]

$\mathbf{h}^{[k]}$: vector for the $k$th hidden layer

$i(z,t)$: water flux boundary condition

$K$: hydraulic conductivity [L T$^{-1}$]





$K_s$: saturated hydraulic conductivity [L T$^{-1}$]

$l$: tortuosity parameter [-]

$L$: the number of hidden layers

$\mathcal{L}$: loss function

$\mathcal{L}_i$: loss term for $i$ constraints

$m$: a van-Genuchten parameter

$n^{[k]}$: dimension of a vector corresponding to the $k$th hidden layer

$n_{VG}$: van-Genuchten parameter [-]

$n^x$: dimension of input vector $\mathbf{x}$

$n^y$: dimension of output vector $\hat{\mathbf{y}}$

$N_i$: the number of points for $i$ constraints

$\mathcal{N}$: neural network functions

$o$: output functions

$q$: water flux in the vertical direction (positive downward) [L T$^{-1}$]

$\mathbf{q}$: water flux in three dimensions [L T$^{-1}$]

$q_A$: constant water flux at the surface to determine the initial condition of the analytical solutions [L T$^{-1}$]

$q_B$: constant water flux at the surface used in the analytical solutions [L T$^{-1}$]

$\hat{r}$: the residual of the RRE

$s$: fixed scaling factor for adaptive activation functions

$S$: source term [T$^{-1}$]

$t$: time [T]

$T$: final time [T]

$\mathbf{W}$: the collection of the weight matrices for a neural network

$\mathbf{W}^{[k]}$: weight matrix for the $k$th hidden layer

$\mathbf{x}$: input vector

$\hat{\mathbf{y}}$: output vector

$z$: vertical coordinate or elevation head (positive upward) [L]

$Z$: the vertical length of a soil [L]


**Greek alphabet**

$\alpha_G$: pore-size distribution parameter [L$^{-1}$]

$\alpha_{VG}$: van-Genuchten parameter [L$^{-1}$]

$\beta$: fixed parameter for the output of neural networks

$\epsilon^\theta$: relative squared error in terms of volumetric water content

$\theta$: volumetric water content [L$^3$ L$^{-3}$]

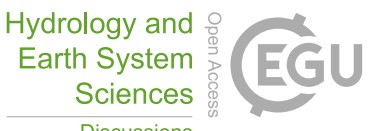

$\theta_r$: residual volumetric water content [$L^3 L^{-3}$]

$\theta_s$: saturated volumetric water content [$L^3 L^{-3}$]

$\Theta$: neural network parameters

$\hat{\Theta}$: update of neural network parameters for each iteration of optimization algorithms

$\kappa_n$: infinite sequence to compute the analytical solution

$\lambda_i$: weight parameters in the loss function

$\sigma$: element-wise non-linear activation function

$\psi$: water potential in soils [L]

$\psi_{lb}$: water potential at the bottom boundary [L]

$\Omega$: spatial domain

$\partial\Omega$: spatial boundary

**others**

:=: Equal by definition

$\nabla$: Nabla

*Author contributions.* TB contributed to the conceptualization, data curation, formal analysis, investigation, methodology, software, valida-tion, visualization, original draft preparation, and review and editing. TG contributed to the conceptualization, funding acquisition, method-
ology, supervision, visualization, review and editing.

*Competing interests.* The authors declare that they have no conflict of interest.

*Acknowledgements.* The publicly available code of physics- informed neural networks provided by Sifan Wang, Yujun Teng, and Dr. Paris Perdikaris (University of Pennsylvania) was instrumental in the development of our model. We are indebted to the editor and anonymous reviewers for improving this manuscript.





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
