# Peer review of "Forward and inverse modeling of water flow in unsaturated soils with discontinuous hydraulic conductivities using physics-informed neural networks with domain decomposition"

_Hydrology and Earth System Sciences, 2022_

## Author Comment (AC1)

Interactive comments on "Forward and inverse modeling of water flow in unsaturated soils with discontinuous hydraulic conductivities using physics-informed neural networks with domain decomposition" by Toshiyuki Bandai and Teamrat A. Ghezzehei

Reviewer comments are in black, while reply to comments in red.

Comments on: Forward and inverse modelling of water flow in unsaturated soils with discontinuous hydraulic conductivities using physics-informed neural networks with domain decomposition.

Summary

This paper presents the results from a comprehensive study using PINNs as a forward and inverse numerical solution for the Richardson-Richards equation. They tested new approaches for applying the PINN method, including a layer-wise locally adaptive function intended to work with layered heterogeneous soil profiles. In addition, the authors compared their approach to well-known numerical solutions for the Richardson-Richards equation, namely Finite Difference and Finite Element Methods (FDM and FEM). The PINNs approach was also validated with soil moisture measurements performed in a soil column in controlled conditions.

The paper appears to be relatively novel, being the first application of PINNs to the Richardson-Richards equation (to my knowledge). The literature proposed and the figures presented are of high quality. I enjoyed reading the paper. The results are encouraging on the applications of PINNs to model hydrodynamics in porous media, even if it takes much more time when compared with the classical approaches. We don't need to impose well-known boundaries and initial conditions, which is attractive once they are difficult to obtain in field applications. The domain decomposition for the layered soils is also very promising. Even classical approaches such as FDM and FEM struggle with heterogeneous soil profiles. So, I think the PINNs with the domain decomposition did quite well in modelling the soil water dynamics in the soil column.

Response: We appreciate you spending time reading our preprint and giving us feedback. We would like to provide answers to your comments and questions.

Specific comments/questions (that should be addressed and commented before publication):

It would be interesting to test the inverse solution with soil matric potential measurements (data is available if needed).

Response: Thank you for pointing out the possibility of using water potential as data. It is not difficult to modify our codes to test the inverse solution with water potential measurements. In the study, we preferred volumetric water content over water potential because volumetric water content sensors are more reliable and cover a wider range of soil moisture conditions (as stated in lines 184-186). We conducted additional numerical experiments using water potential data from the same HYDRUS-1D simulation used in the inverse modeling in the main text, and the estimated surface water flux is comparable to the case using volumetric water content (see Figure 1 below). We will include this result

in the supplementary material in the next revision.

[Figure]

*Figure 1. Inverse modeling to estimate surface flux from five water potential measurements in a layered soil ($z \in \{-1, -5, -9, -13, -17\}$ cm). The left figure shows the comparison between the true and PINNs' volumetric water content. The right figure shows the true and estimated surface water flux.*

What is your opinion on going to 2 and 3D modelling? Could the domain decomposition proposed in the paper be applied to speed up 2 and 3D solutions? I think that would be the actual gain in this methodology. FEM applications for the fully 3D solution of Richardson-Richard's equation are still slow and have many complications with mesh, especially for large domains. This also applies to the boundary and initial conditions imposition.

Response: Thank you for your suggestions. Yes, the domain decomposition was indeed invented for speeding up PINNs for large-scale simulations by dividing spatial and temporal domains into smaller ones (Jaqtaq and Karniadakis, 2020). Although we presented results in only 1D to understand how PINNs behave, we are moving toward 2D and 3D simulations using PINNs.

What about non-Darcian conditions, macropore flow, very high clay content soils. Do you think the method could be applied?

Response: We appreciate your comments on those processes. A simple answer is "as long as we can describe those processes as mathematical equations, we should be able to simulate those processes using PINNs." Please think of PINNs as numerical solvers, such as finite difference and finite element methods.

The mathematical formulation of non-Darcy flow was proposed by Swartzendruber (1962), for instance, and this formulation can be implemented using PINNs. As for macropore flow, we believe we do not have a good mathematical model that can take into account the effects of macropore flow on the overall water flow (Nimmo, 2021). Thus, it is difficult to use PINNs for macropore simulation (same for traditional finite difference and finite element).

In terms of water flow in very high clay content soils (i.e., swelling soils), we believe we can simulate water flow in clay-rich soils that can shrink and swell. J. Philips and D. E. Smiles studied infiltration into swelling soils and provided the mathematical formulation of the processes (e.g., Smiles, 1974). Note that the water flow rate would be flow rate "relative to soil particles" in that case, and we need to take into account the change in porosity. Although these simulations are not common, such mathematical models exist, and thus we think we can apply PINNs to clay-rich soils.

What about root-water-uptake? How can this be included in your approach? There exist some analytical solutions for these problems (Yuan and Lu, 2005[1])

Response: I appreciate your comment and the literature you suggested. The answer to this question would be the same as the one before. As long as we have mathematical models for root-water-uptake, we can use PINNs to simulate as we do using finite difference and finite element methods. Since we have some root-water-uptake models (e.g., Feddes and Raats, 2004), we can include the plant-root water uptake as a sink term in the Richards equation and implement PINNs.

Do you think one day the PINNs could take over the classical approaches? What is limiting it?

Response: This is an important question. We do not see PINNs taking over the classical approaches in the future. We instead envision combining PINNs with classical approaches. For example, we can use a finite element solution for large-scale simulation with a coarse mesh size to train PINNs and later decrease "mesh size" (using more residual points) to get more refined solutions using PINNs, where the finite element method cannot be used due to a significant amount of degree of freedoms. The limitation of PINNs is the difficulty in training PINNs. However, we recently observed an exciting breakthrough in training PINNs, so we expect training PINNs to be more efficient and consistent in the future.

What about practical applications? Irrigation management or contaminant transport in the vadose zone.

Response: Thank you for commenting on practical applications. Current practical applications are to estimate rainfall estimations from soil moisture measurements (directly related to the inverse problem shown in the main text). As for irrigation management, we can formulate an inverse problem for irrigation management, where "desired soil moisture distribution" would be used to train PINNs to determine "required irrigation to achieve the desired soil moisture distribution." We might be able to use PINNs to locate the source of contaminant from measured contaminant data in the vadose zone by solving an inverse problem, where a sink term in a convective-dispersion equation is to be estimated. We would like to emphasize that those problems are all inverse problems, so PINNs and traditional approaches are both applicable.

Overall, the paper is well written. The sections are balanced, and the flow is good, making the paper enjoyable to read.

Response: We thank your constructive comments and feedback. We will include some of the answers here in the revised manuscript.

References

Feddes, R. A., & Raats, P. A. C. (2004). Parameterizing the soil - water - plant root system. In R. A. Feddes, G. H. de Rooji, & J. C. van Dam (Eds.), Unsaturated-zone modeling: Progress, challenges, and applications (pp. 95–141). Springer Netherlands.

Jagtap, A. D., & Karniadakis, G. E. (2020). Extended physics-informed neural networks (XPINNs): A generalized space-time domain decomposition based deep learning framework for nonlinear partial differential equations. Communications in Computational Physics, 28(5), 2002–2041. https://doi.org/10.4208/cicp.oa-2020-0164

Nimmo, J. R. (2021). The processes of preferential flow in the unsaturated zone. Soil Science Society of America Journal, 85(1), 1–27. https://doi.org/10.1002/saj2.20143

Smiles, D. E. (1974). Infiltration into a swelling material.pdf. Soil Science, 117(3), 140–147.

Swartzendruber, D. (1962). Modification of Darcy's law for the flow of water in soils. Soil Science, 93(1), 22–29.

---

## Author Comment (AC2)

Interactive comments on "Forward and inverse modeling of water flow in unsaturated soils with discontinuous hydraulic conductivities using physics-informed neural networks with domain decomposition" by Toshiyuki Bandai and Teamrat A. Ghezzehei

Reviewer comments are in black, while reply to comments in red.

Review of "Forward and inverse modeling of water flow in unsaturated soils with discontinuous hydraulic conductivities using physics-informed neural networks with domain decomposition" by Bandai and Ghezzehei.

In this manuscript the authors tested a physics-informed neural networks (PINNs) method to solve the Richardson-Richards equation for simulating unsaturated soil water dynamics. The authors also investigated the capability of the method for obtaining inverse solutions. As coupling data-driven and physics-based approaches have received much attention these days, the topic fits well with the scope of HESS. The authors have done a great job on demonstrating how PINNs performed when simulating unsaturated water flow in soils and showing applicability and limits of the method. Although the paper was well organized and written, I believe the paper has a room for some improvement. I have some comments that should be addressed prior to accepting this paper for publication. For my curiosity, I am wondering if this approach can be applied to simulate preferential type flow in soils. Is it going to be straightforward? Does it require some modifications in the model? If it can be applied to such phenomena, it would be a great breakthrough in the field of soil physics and hydrology.

Response: We sincerely thank the reviewer for spending efforts in reviewing our manuscript. Before we answer the questions below, we would like to clarify the application of PINNs to preferential flow here. Although it is highly important to simulate preferential flow, the current PINNs cannot be applied to general preferential flows. This is because reliable mathematical models for preferential flow have not been developed yet. The basis of PINNs is well-defined mathematical equations (e.g., differential equations) that describe the processes of interest. This is an identical requirement to other traditional numerical methods such as finite difference and finite element methods. Nevertheless, some aspects of preferential flow could be simulated using PINNs. For example, Cueto-Felgueroso and Juanes (2009) proposed to model finger flow in a homogeneous soil by adding a fourth-order term to the Richardson-Richards equation. The fourth term describes the formation of gravity fingers during water infiltration into soils. In principle, we can apply PINNs to solve the fourth-order Richardson-Richards equation, though it requires further computational cost due to the fourth-order term. Alternatively, it might be possible for us to use PINNs to study the gravity finger using laboratory infiltration experiments using imaging-based soil moisture data (e.g., Sadeghi et al. (2017)). Compared to traditional numerical methods, PINNs do not require initial and boundary conditions, which would be useful for the experimental laboratory setup too. In conclusion, PINNs are limited to processes that mathematical equations are available for. We require more experimental and theoretical work on formulating mathematical models for preferential flow in general (Nimmo, 2021).

General comments:

In Fig. 5, the evolution of PINNs solution is plotted. At the initialization, some of the solutions are beyond the limit of the water content as the water content values are greater than the saturated water content. Would it be possible to put some constraints to the solutions in PINNs? If so, would that improve training and overall performance? Any discussions on this matter will helpful for those who are interested in using this method. A similar question goes to the inverse solutions. I am wondering if any constraints can be applied to the target parameters that are inversely estimated. There is always a need to put some constrains to the parameters being estimated.

Response: We appreciate your keen comments on Figure 5. The oversaturation at the initialization is because we did not implement the conditional statement such as $\theta = \theta_s$ for $\psi \geq 0$. Thus, the neural networks gave unphysical volumetric water content over the saturated water content based on the Gardner model (Equation (5)).

We had some technical issues that prevented us from implementing the conditional statement. But, we have now implemented such conditional statements. An example of the results from this implementation is shown below in addressing your questions on saturated-unsaturated flow.

In terms of your question on the constraint on inverse modeling in general, yes, we can constrain the range of target variables in this framework. For example, if the range of a variable $\alpha$ is $0 < \alpha < 1$, then we can use the sigmoid function to represent $\alpha$ (note that the range of the sigmoid function is between 0 to 1). Alternatively, we might set bounds for each parameter by adding conditional statements. Nevertheless, as standard inverse modeling frameworks, imposing such bound constraints might make the minimization problem more difficult than that for unconstrained optimization.

In the demonstration of getting inverse solution with PINNs, the authors used a 2-layered soil system. Why? If the boundary fluxes are being estimated, wouldn't be better to start with a homogenous case? Was there a specific reason that the layered soil system was used in this demonstration?

Response: We appreciate your reasonable suggestion. The reason we used a two-layered soil for the inverse modeling is twofold: 1) we knew PINNs work well for a homogeneous soil from our previous study (Bandai and Ghezzehei, 2021); 2) soil moisture sensors are inserted into multiple layers in our target field data, which is often the case for most situations. We believe it is important to test this algorithm for two-layered soils because the physical property of the very surface soil is often different from the below ones due to crust, organic matter accumulation, and surface processes.

Specific comments:

L189: It sounds a bit strange to say that soil dynamics is "controlled by the volumetric water content at the bottom."

Response: Thank you for pointing out this. To avoid confusion, we will change the sentence:

"which corresponds to when soil moisture dynamics is controlled by the surface water flux q(0, t) (i.e., evaporation or infiltration) and the volumetric water content at the bottom θ(−Z, t)."

to

"which corresponds to when soil moisture dynamics is induced by the surface water flux q(0, t) (i.e., evaporation or infiltration) while the volumetric water content at the bottom θ(−Z, t) is kept to h(-Z, t)."

L193 (Eq.15): A little bit more explanations will be helpful to understand this transformation. I have no idea why the beta value gives better initial guess.

Response: We appreciate your comment. The initial guess for the homogeneous simulation is shown in Figure 5 (a). This distribution is determined by Equation 15 with the parameter $\beta$ and the initial neural network parameters Θ. The initial neural network parameters are given by the Glorot initialization (Line 262). By tuning the parameter $\beta$, we can begin our simulation at a better initialization. The parameter $\beta$ is also important for the question below (L311).

L311: If the logarithmic transformation of water potential is used, the approach is limited to "unsaturated" systems. But there are many cases you will have both positive and negative potential values. How do you deal with that?

Response: Thank you for your insightful comment. As you suggested, the output of the negative logarithmic transformation is always negative. However, the parameter $\beta$ in Equation (15) makes the PINNs possible to have positive water potential (i.e., saturated flow). For example, if we use $\beta = 10$, the range of $\hat{\psi}$ would be $-\infty < \hat{\psi} < 10$.

Although we limited ourselves to unsaturated cases in this paper, we tested the applicability of PINNs to saturated-unsaturated cases by changing the surface flux of the homogeneous problem in the main text from 0.9 cm/h to 1.05 cm/h, where the saturated hydraulic conductivity is 1.0 cm/h. Figure 1 below shows that the volumetric water content reached the saturated volumetric water content at t = 3 h for half of the soil, while the water potential becomes positive, as shown in Figure 2. The slope of the water potential is close to 0.05 cm/cm, which is reasonable to satisfy the surface flux to be 1.05 times the saturated hydraulic conductivity.

[Figure]

*Figure 1. The PINNs solution for saturated-unsaturated flow into a homogeneous soil in terms of volumetric water content.*

[Figure]

*Figure 2. The PINNs solution for saturated-unsaturated flow into a homogeneous soil in terms of water potential.*

We sincerely appreciate you gave us a chance to test PINNs against the saturated-unsaturated flow. However, we can foresee the potential difficulty of PINNs for saturated-unsaturated cases because the solution to the governing equation will not be smooth (i.e., not differentiable) at the saturated-unsaturated interface, and it is difficult to know the location of the interface a priori. Please note that traditional numerical methods have also issues for saturated-unsaturated interfaces (convergency of non-linear solver). The application of PINNs to saturated-unsaturated flow requires further verification and comparison with other numerical methods, and we would like to limit ourselves to unsaturated scenarios in this paper, as suggested in the title. We will add these comments in the discussion section of the revised manuscript as a future perspective.

Figure 3(b): There are some systematic differences between FDM and PINNs. Why? Are these because of the choice of spatial and temporal discretization in FDM?

Response: Thank you for pointing out the systematic difference. Yes, the difference comes from the discretization used in FDM. In FDM, the temporal derivative is approximated by a finite difference, and the FDM solution and the corresponding numerical error propagate with time marching. Major errors come from a steep wetting front, and thus we see a clear trend in FDM errors. On the other hand, PINNs do not use time marching for temporal discretization and minimize the residual of the partial differential equations in both time and space simultaneously. Therefore, we see PINNs errors distributing broadly in the spatial and temporal domains.

Figure 10: Looks like something is wrong with the texts at the top of the figures.

Response: We regret that we left the weird texts in the figure. We will fix this in the revised manuscript.

Figure 16: For all three cases, the PINN solutions show that the inversely estimated initial surface flux is much smaller than the true flux. Are there any specific reasons for this?

Response: We appreciate your close investigation. According to our empirical experiences, PINNs tend to learn the solution backward in time. Please see Figure 13, where PINNs learned the solution at t = 10 h first and other solutions backward in time. We believe the same thing happened to the inverse modeling case too. Please see Figure 15 (a), where the solutions at t = 0 and t = 1 are not satisfactory near the surface. The estimated water content at t = 0 and t = 1 would

be closer to the solution at t = 3 because neural networks are continuous, and thus estimated water content is higher than the true water content. If the estimated water content is higher than the true one, then there is less water flux required, which is the result of the underestimation (in absolute value) of initial surface water flux.

References

Bandai, T., & Ghezzehei, T. A. (2020). Physics-informed neural networks with monotonicity constraints for Richardson-Richards equation: Estimation of constitutive relationships and soil water flux density from volumetric water content measurements. Water Resources Research, 57, e2020WR027642. https://doi.org/10.1029/2020WR027642

Cueto-Felgueroso, L., & Juanes, R. (2009). A phase field model of unsaturated flow. Water Resources Research, 45(10). https://doi.org/10.1029/2009WR007945

Nimmo, J. R. (2021). The processes of preferential flow in the unsaturated zone. Soil Science Society of

America Journal, 85(1), 1–27. https://doi.org/10.1002/saj2.20143

Sadeghi, M., Sheng, W., Babaeian, E., Tuller, M., & Jones, S. B. (2017). High‐resolution hortwave infrared imaging of water infiltration into dry soil. Vadose Zone Journal, 16(13). https://doi.org/10.2136/vzj2017.09.0167

---

## Author Comment (AC5)

Interactive comments on "Forward and inverse modeling of water flow in unsaturated soils with discontinuous hydraulic conductivities using physics-informed neural networks with domain decomposition" by Toshiyuki Bandai and Teamrat A. Ghezzehei

Reviewer comments are in black, while reply to comments in red.

The paper is very interesting and introduces a physics-informed neural networks (PINNs) method in a Richards' equation context, aimed at approximating the solution to the RRE using neural networks while concurrently matching available soil moisture data. In particular, in this paper authors consider domain decomposition for handling infiltration into layered soils.

The topic is definitely up to date, the paper is well written, and provides all the details for implementing and understanding this approach. Nevertheless, I think authors should address some comments and issues before it can be accepted.

Response: We appreciate that the reviewer closely read our manuscript and gave the valuable comments. We would like to answer the questions below.

This PINN approach appears really fascinating because it allows to integrate physics-based models (such as RRE) with machine learning features. Authors ascribe the uncertainties in Richards'equation to the choice of boundary conditions, which is surely right. Nevertheless, I think they do not consider the (even more) cumbersome uncertainties arising from the choice of model parameters, which are the result of some non-linear fitting in laboratory experiments (I am referring to the parameters in the WRC and HCF). This point is a main concern for me: as a matter of facts, unsaturated flow dynamics strongly relies on functions parameters, rather than on ICs and BCs, which are generally easier to assess. On the other hand, I see authors already published a paper on this topic: I think it would be valuable to stress the differences between the two papers

Response: We appreciate your comments on the concern regarding the soil hydraulic parameters. We deeply acknowledge the effect of the uncertainty of the soil parameters on the soil moisture dynamics. From our previous study (Bandai and Ghezzehei, 2021), where we attempted to estimate the parameters from soil moisture measurements, we recognized the importance of understanding the basic characteristics of PINNs constrained by the RRE and its extension to layered soils for practical applications. In the revised manuscript, we will stress this point. In future studies, we will include the uncertainty of the parameters into this PINNs framework again.

Lines 197-198: few more words for sketching how the partial derivatives are computed would be valuable

Response: Thank you for pointing out that there needs more explanation on the automatic differentiation. In this method, all the computations necessary for PINNs are formulated as computational graphs by the software (TensorFlow in this study). Any derivatives related to the computations can be computed based on the reverse-mode automatic differentiation (basically, chain rule). The cost of computing the derivative is similar to or less than conducting the computation twice, regardless of the number of parameters. Therefore, this method is suitable for training neural networks with tens of thousands or millions of parameters. We will add an explanation of automatic differentiation in the revised manuscript.

Figure 1: I think there is a typo in the box "Physics and Data Constraints", since the partial derivative at the left-hand side should be accomplished with respect to time.

Response: We regret to have left the typo in the figure. Thank you very much for noticing the mistake. We will fix this in the revised manuscript.

I understand that the residual is computed between the synthetic data and the computed (by the PINNs) ones; in this framework, what is the rationale of comparing the PINNs output with any Richards solver (as Hydrus)?

Response: The comparison with other RRE solvers (such as HYDRUS-1D) was conducted in the forward modeling, where we verified the ability of PINNs to approximate the solution to the RRE. It is important to make sure the performance of PINNs for the forward modeling because the performance of PINNs for the inverse modeling partially depends on the performance of the forward modeling.

As far as I understand, the power of this approach is to combine physics-based models with data driven ones; according to my knowledge, this is also the spirit of Data Assimilation (DA) methods, which incorporate measurements into a physics based model, albeit in a very different framework; these methods have also been treated in Richards' equation context (see for instance Berardi et al CPC https://doi.org/10.1016/j.cpc.2016.07.025, Medina et al HESS https://doi.org/10.5194/hess-18-2521-2014, Liu et al JoH https://doi.org/10.1016/j.jhydrol.2020.125210 ); what is authors' opinion about this? What would be the pros and cons of PINNs approach with respect to DA one? Also DA methods allow to assimilate boundary conditions, as in this case, and hydraulic parameters, as well as states. As a matter of fact, with respect to DA methods, this PINNs approach seems to me more on the theoretical side (which is definitely fine, of course) rather than application oriented.

Response: We appreciate that the reviewer pointed out the relation with data assimilation (DA). Our view is that both PINNs and DA are methods for inverse problems, where we aim to extract information from measurement data (Asch et al., 2016). We answer the questions by dividing DA methods into 3-D or 4-D variational methods (also called adjoint methods) and Kalman filter methods.

Adjoint methods are closely related to PINNs. In fact, we are currently working on the comparison between PINNs and adjoint methods. The difference comes from, for instance, the basis functions (linear functions for adjoint methods and neural networks for PINNs) and the method of minimizing the objective function. However, the methods might give very similar results based on our experiences. We would like to report this finding in future studies.

As the reviewer commented, the Kalman filter method might be more practical because it can also give an uncertainty of the results in addition to the point estimate. To the best of our knowledge, we are not aware of the comparison between the Kalman filter method and PINNs.

To conclude, our opinion is that all the methods mentioned here are within inverse modeling. The efficacy of each method would depend on the system of interest and the available data. As a research community, we need some benchmark problems to compare the methods.

Authors mention the possibility to drop loss terms for IC or BC at line 225. However, they have not presented any experiment for this scenario. Could you please comment on this ill-posed configuration? How would it perform with respect to classical solver?

Response: It might have been unclear in the current manuscript. The ill-posed setting was used for the inverse modeling, where no initial and boundary conditions were enforced, and only measurement data were used to train PINNs. In terms of classical solvers (finite difference or finite element), it is also possible to deal with such ill-posed settings by treating the missing information (e.g., initial condition) as inversion parameters and formulating the problem as inverse modeling. This type of inverse modeling is hard to implement in standard hydrology software (e.g., HYDRUS-1D) because the number of parameters can be very large. However, in geophysics or optimal control fields, it is common to deal with this inverse problem using adjoint methods (e.g., Petra and Stadler, 2011; we will add this literature to the revised manuscript).

Figure 11 and 3. Please replace "Fintie" with "Finite".

Response: Thank you very much for noticing the mistake. We will fix this in the revised manuscript.

Authors make use of synthetic data: I had hard times to find where the reference to used data is described. Of course the use of synthetic data is fine, but they should highlight it at the beginning of the paper. Moreover, could you please explain how your method of synthetic data generation could compare to real measurement data? In other words, how robust is your result with respect to outliers, sensor noise and other technical issues when it comes to real data?

Response: We appreciate that the reviewer pointed out that it was not clear if the data are synthetic in the current manuscript. We will emphasize that regard in the introduction of the revised manuscript. The robustness of the algorithm against the sensor noise was addressed by incorporating the Gaussian noise into the synthetic data used in the inverse modeling section (Line 559-560). The other technical difficulties (e.g., outliers and model errors) were not addressed in this manuscript. In the real setting, outliers should be removed and not be fed into the algorithm because the algorithm assumes soil moisture dynamics can be described by the Richardson-Richards equation. As for model errors, as long as they can be described by the Gaussian distribution, they can be interpreted as the Gaussian noise as in the current manuscript. For example, we did not consider the effect of hysteresis on soil moisture dynamics in this study. The model error due to hysteresis might be described by the Gaussian noise for wetting and drying situations. However, If there are biases that cannot be described by the Gaussian in the model error, we need to update the model by incorporating those processes as forms of mathematical equations.

References

Asch, M., Bocquet, M., Nodet, M. (2016). Data assimilation: methods, algorithms, and applications. Fundamentals of Algorithms. SIAM.

Bandai, T., & Ghezzehei, T. A. (2021). Physics-informed neural networks with monotonicity constraints for Richardson-Richards equation: Estimation of constitutive relationships and soil water flux density from volumetric water content measurements. Water Resources Research, 57, e2020WR027642. https://doi.org/10.1029/2020WR027642

Petra, N., & Stadler, G. (2011). Model variational inverse problems governed by partial differential equations. In ICES Report.